

# Antarctic subglacial lakes drain through sediment-floored canals: Theory and model testing on real and idealized domains

Sasha. P. Carter[1], Helen. A. Fricker[1], and Matthew. R. Siegfried[1]

[1]Institute of Geophysics and Planetary Physics, Scripps Institution of Oceanography, University of California, San Diego, CA, USA

*Correspondence to:* S. P. Carter (spcarter@ucsd.edu)

**Abstract.** Over the past decade, satellite observations of ice surface height have revealed that active subglacial lake systems are widespread under the Antarctic ice sheet, including the ice streams. For some of these systems, additional observations of ice stream motion have shown that lake activity can affect ice-stream dynamics. Despite all this new information, we still have insufficient understanding of the lake-drainage process to incorporate it into ice sheet models. Process models for drainage of

ice-dammed lakes based on conventional "R-channels" incised into the base of the ice through melting are unable to reproduce the timing and magnitude of drainage from Antarctic subglacial lakes estimated from satellite altimetry given the low hydraulic gradients along which such lakes drain. We have developed an alternative process model, in which channels are mechanically eroded into the underlying deformable subglacial sediment. When applied to the known active lakes of the Whillans/Mercer ice stream system, the model successfully reproduced both the inferred magnitudes and recurrence intervals of lake volume

changes, derived from Ice, Cloud and land Elevation Satellite (ICESat) laser altimeter data for the period 2003–2009. Water pressures in our model changed as the flood evolved: during drainage, water pressures initially increased as water flowed out of the lake primarily via a distributed system, then decreased as the channelized system grew, establishing a pressure gradient that drew water away from the distributed system. This evolution of the drainage system can result in the observed internal variability of ice flow over time. If we are correct that active subglacial lakes drain through canals in the sediment, this

mechanism also implies that active lakes are typically located in regions underlain by thick subglacial sediment, which may explain why they are not readily observed using radio-echo sounding techniques.

## 1 Introduction

Since the initial observation of "large flat circular basins" on the ice surface of Antarctica by Russian pilots during the International Geophysical Year (Robinson, 1964), there has been interest on the role of lakes within the larger ice sheet system.

Beginning in 1972, radio-echo sounding (RES) began confirming that these surface features reflect storage of free water at the ice-sheet base (Oswald and Robin, 1973) and subsequently continued to be a primary technique to identify subglacial lakes, including Lake Vostok, one of the largest freshwater lakes in the world (Ridley et al., 1993; Kapitsa et al., 1996). With increased availability of RES data and as our ability to precisely observe the ice surface has improved, the number of known subglacial lakes in Antarctica has increased. At the time of writing, over 300 subglacial lakes have been discovered throughout the conti-



nent using a variety of geophysical methods (Wright and Siegert, 2012). Until the mid 2000s radio-echo sounding (RES) was the primary technique for identifying subglacial lakes (e.g. Siegert et al., 2005; Carter et al., 2007). Most of the lakes found in RES surveys tended to be located beneath the slow-moving ice near the divides (Figure 1a), driving initial research questions on whether lakes were open or closed systems (e.g. Bell et al., 2002; Tikku et al., 2005), with considerable speculation about

their impact on local ice dynamics (e.g. Dowdeswell and Siegert, 1999; Bell et al., 2007; Thoma et al., 2012).

Since 2005, a variety of repeat observations of the ice surface from satellite missions have revealed patterns of surface uplift and subsidence consistent with the filling and draining of subglacial water bodies (e.g. Gray et al., 2005; Wingham et al., 2006; Fricker et al., 2007). In contrast to "RES lakes", these "active lakes" have been found beneath fast flowing ice streams and outlet glaciers (Figure 1a, 1b). Many of the active lakes that have been surveyed with RES (e.g. Christianson et al., 2012;

Siegert et al., 2014; Wright et al., 2014) lacked the characteristic basal reflections traditionally used to identify the presence of subglacial lakes (hydraulic flatness, specularity, and brightness relative to surroundings; see Carter et al. (2007)). Multiple hypotheses have been proposed to explain this discrepancy, mostly relating to unconstrained hydrodynamics of the system or data processing artifacts (Siegert et al., 2015).

Active lakes are of particular glaciological interest due to their potential impact on the dynamics of fast-flowing ice streams.

Surface altimetry observations suggest that these lakes can hold back and then episodically release large volumes of water into the subglacial environment (e.g. Fricker et al., 2007; Smith et al., 2009; Carter et al., 2011), which could alter ice-stream dynamics for 100s of km downstream. Repeat measurements of ice velocity coincident with the existing record of lake dynamics are sparse, with only three published instances: Byrd Glacier, East Antarctica (Stearns et al., 2008); Crane Glacier, Antarctic Peninsula (Scambos et al., 2011); and Whillans Ice Stream, West Antarctica (Siegfried et al., 2016). In all three

cases, subglacial lake activity inferred from surface height anomalies correlated to episodes of temporary ice acceleration, but understanding of longer term impact of lake drainage on ice dynamics is significantly limited by a short (maximum 12-year) observational window (Siegfried et al., 2016).

Critical to resolving the link between ice dynamics and lake activity is determination of the mechanism by which lake drainage occurs. While some ice sheet models have started to incorporate primitive elements of subglacial lake dynamics (e.g.

Goeller et al., 2013; Livingstone et al., 2013), they still do not have a realistic treatment of observed lake drainage processes. Most process-based treatments of Antarctic subglacial lakes (e.g. Wingham et al., 2006; Evatt et al., 2006; Carter et al., 2009; Peters et al., 2009) have hypothesized that active lakes drain via a mechanism similar to that of ice-dammed lakes in temperate glacial environments, in which narrow, semi-circular conduits are melted into the basal ice ("R-channels"). More recently, however, several complimentary lines of evidence have called into question the ability of an R-channel to form (Hooke and

Fastook, 2007) and close (Fowler, 2009) in subglacial conditions typical of Antarctica. Fowler (2009) suggested that channels incised into the underlying sediment may be the preferred mechanism.

In this paper we describe the development of a new model for the filling and drainage of Antarctic subglacial lakes on ideal and real domains based upon several well-studied lakes in the hydrological system of Mercer and Whillans ice streams, West Antarctica (Fricker et al., 2007; Fricker and Scambos, 2009; Siegfried et al., 2014; Tulaczyk et al., 2014; Siegfried et al., 2016).

Lake drainage occurs in our model through channels in the underlying sediment (originally termed "canals" by Ng, 2000) and



we compare the output lake volumes with those from an existing R-channel model (following Kingslake and Ng, 2013). We aim to: (i) develop a lake drainage model that reproduces the recurrence interval and inferred volume ranges for lake drainage events in the Siple Coast; (ii) provide better context for ongoing observations of lake volume change and lake distribution with respect to the subglacial hydrology; and (iii) move towards a consistent parameterization of subglacial lake activity in ice sheet

models.

This paper begins with background on existing subglacial water models, both for Antarctica and for ice-dammed lakes in temperate environments (Sect. 2). Next, we describe the theory of the models used in this manuscript, including background hydropotential theory, our implementations of R-channel and canal theory, and our coupling between a background distributed water system and a channelized system. We also detail the domains over which we apply the models (Sect. 3). We then highlight

the results from our experiments using the R-channel and canal models, compare these results to published observations of Antarctic subglacial drainage events, and explore the sensitivity of our model to various physical parameters required as model inputs (Sect. 4). We discuss the implications of these model sensitivities, suggest reinterpretations for existing observations of subglacial hydrology in Antarctica based on the viability of a canal mechanism for lake drainage, and consider issues related to the inclusion of realistic models of subglacial lake drainage into large-scale ice sheet models (Sect. 5). Finally, we summarize

our findings (Sect. 6).

## 2   Basal water models and subglacial lake drainage

### 2.1   Antarctic basal water models

Models for subglacial water transport and distribution include at least one of three processes: distributed sheet flow; groundwater; and channelized flow. The simplest and most common models for ice sheet basal water flow invoke some form of

distributed system that spreads water laterally (e.g. Le Brocq et al., 2009). In such systems, water pressure increases with water flux, while basal traction decreases. More sophisticated models, however, prefer to accommodate sliding by deformation of the subglacial sediment, making basal traction decrease through increasing sediment porosity (e.g. Tulaczyk et al., 2000). Given that subglacial sediment is widely understood to lack the transmissivity necessary to accommodate the water fluxes at the base of the Antarctic ice sheet (Alley, 1989), changes in sediment water storage have been regulated by exchange with a distributed

system (e.g. Christoffersen et al., 2014; Bougamont et al., 2015). These more sophisticated distributed / groundwater exchange models show the most consistency with borehole (Engelhardt and Kamb, 1997; Christner et al., 2014; Tulaczyk et al., 2014) and seismic (Blankenship et al., 1987) observations of the basal environment.

Channelized systems are those in which water flux is concentrated in one or more discrete conduits. The type of conduit that is most commonly modelled is referred to as an "R-channel", which is thermally-eroded into the ice by turbulent heat

generated by water moving down a hydraulic gradient (Röthlisberger, 1972; Nye, 1976). As the relative area of the ice-bed interface occupied by these systems is small, they can support lower water pressures. R-channels have been well studied in Greenland, where they have been directly observed at the ice sheet margin, and are understood to be supplied primarily by surface water, much of which enters at discrete recharge points known as moulins (e.g. Zwally et al., 2002). Basal meltwater



systems in Antarctica differ in two significant ways from those in Greenland: they are isolated from the atmosphere and so do not receive surface meltwater; and they are located in regions of low hydraulic slopes, such that the heat generated by water moving down gradient is likely not sufficient to erode an R-channel (Alley, 1989). Therefore, channelization has typically not been considered in large-scale basal water models for Antarctica (e.g. Le Brocq et al., 2009; Bougamont et al., 2011). In the

last decade, however, increased consideration has been given to the role of channelization in the drainage of subglacial lakes (e.g. Evatt et al., 2006; Carter et al., 2009), and to the possibility of subglacial channels incised in the sediments instead of the overlying ice (e.g. van der Wel et al., 2013; Kyrke-Smith and Fowler, 2014).

## 2.2 Lake-drainage theory

Most of our understanding of the drainage of subglacial lakes derives from the Nye (1976) model for the drainage of glacial-

dammed lakes on temperate glaciers in alpine environments (e.g. Clarke, 2003; Werder et al., 2013) where floods descend 100s of meters over 1s to 10s of km on timescale of days and channelization is well documented (e.g. Roberts, 2005). Fowler (1999) explained the repeated drainage in these systems with the following model (see also Figure 2): (i) when the lake is at low stand, water is trapped behind a local maximum in hydropotential, known as "the seal"; (ii) as lake water levels rise, a hydraulic connection forms over the seal initiating thermal erosion by outflow from the lake; (iii) during the early stages of lake

drainage, the potential gradient is relatively steep and effective pressure is low causing melt to exceed creep closure; (iv) with ongoing drainage, the level of the lake lowers causing a decrease in the hydraulic gradient, while the effective pressure at the seal increases, allowing the channel to continue to siphon outflow despite the lower lake-level. The effective pressure change also increases the rate of creep closure of the channel by the overlying ice, while the energy available for thermal erosion simultaneously decreases; (v) as the channel closure rate increases, a reduction in effective pressure ultimately re-forms a

hydraulic seal between the lake and points downstream.

Antarctic subglacial floods occur on larger spatial and longer temporal scales than alpine subglacial floods: water typically descends 10s of meters over 100s of km and drainage sometimes persists for multiple years (Wingham et al., 2006; Fricker et al., 2007). Limited evidence from borehole observations of the basal environment along major flow paths connecting lakes indicates that water travels via distributed systems (e.g. Engelhardt and Kamb, 1997). Although the patterns of lake volume and

outflow over time are qualitatively similar to those observed during the drainage of alpine-ice dammed lakes, both historical and more recent work has shown that R-channels are unlikely to play a major role in the Antarctic subglacial water system due to several issues:

1. The inability of water to melt a channel on an adverse bedslope: a substantial number of flow paths that drain known subglacial lakes appear to exist on an adverse bed slope from thicker ice into thinner ice, where the pressure melting

point is higher. Alley et al. (1998) showed that once the basal slope exceeds 1.2–1.7 times the surface slope, the heat generated by turbulent dissipation is insufficient to maintain the water at the pressure melting point and therefore cannot melt surrounding ice to form an R-channel;





2. Thermal issues related to polar ice: most models for R-channel formation imply that the surrounding ice is temperate and isothermal. In polar ice where a significant temperature gradient is likely to be present in the ice immediately above the bed, more turbulent heat will be required to melt a given amount of ice than would be needed for temperate ice (Hooke and Fastook, 2007); and

3. The slow rate of closure predicted for R-channels at pressures observed under Antarctica: creep closure of R-channels requires a drop in water pressure on the order of several 10's of meters water equivalent, 5–10 times higher than the surface drawdown observed during the drainage of most Antarctic subglacial lakes (Fowler, 2009). Field reports suggest that subglacial (SLW) contained water even at low stand (Christianson et al., 2012; Horgan et al., 2012; Christner et al., 2014), suggesting that the channel may have closed before the lake was completely drained.

In our work we directly compared output from a model for lake drainage via an R-channel (Kingslake and Ng, 2013) with a model in which channels are formed via mechanical erosion of underlying sediment (canals) on a domain with geometry similar to that found for flow paths draining Antarctic subglacial lakes. By replacing a channel incised into the overlying ice with one incised into the sediment we address issues (1) and (2), as the erosion of sediment is less temperature sensitive than the erosion of ice. This change also addresses (3) because the deformability of sediment is more sensitive to small changes in water pressure than deformability of ice (Fowler, 2009). We tested the output from these models against observations to determine which model was able to best reproduce estimates of the inferred magnitude and recurrence interval for known subglacial lake drainage events.

## 3    Model description

Our model is a prototype for lake drainage via a single channel incised into a linearly viscous subglacial sediment. The overall principle of lake drainage via a channel where water pressures are significantly below overburden pressure (in our case 2–8 m w.e.) is well established in the literature (e.g. Nye, 1976; Fowler, 1999; Evatt et al., 2006). Only recently has it been suggested that such drainage may occur via channels forming in sediment rather than the overlying ice (e.g. Fowler, 2009). In order to accommodate a channel incised in sediment rather than in ice, we adopted principles of sediment mechanics from theoretical work presented in Walder and Fowler (1994) and Ng (2000).

Given that borehole observations in the region have indicated a distributed system at the ice-bed interface (e.g. Engelhardt, 2004) we needed a simple model for the coupling of the channel to a distributed system. Therefore, we adapted a formulation presented in Kingslake and Ng (2013) which, although designed for a rocky-bottomed mountain glacier lake drainage, contained the four most essential elements we required for our ice-stream experiment: a lake, a distributed flow system, a channelized system and a means for communication between all parts of the system. We developed two models based on the Kingslake and Ng (2013) foundation: one model with channelization through conduits melted into the basal ice based on R-channel theory (henceforth called the R-channel model) and one model with channelization through canals eroded into the basal sediment (henceforth called the canal model). Although there may be more appropriate models for each of the individual



components in the Antarctic subglacial environment (see Flowers (2015) for a review of existing water models), we found the ease of coupling between the lake, channelized system and the distributed system provided by the Kingslake and Ng (2013) formulation to be the most effective for proof of concept. Input data for the model comes from several sources: previous water budget studies for lake inflow estimates, radar sounding for ice thickness, and satellite and airborne radar altimetry for sur-

face height changes. By comparing the comparing lake volume change inferred from satellite observations with lake volume change as simulated with R-channel and canal models, we can (i) perform a diagnostic test for the hypothesis by Fowler (2009) that drainage could occur for sediments that behave like erodible-deformable ice; and (ii) develop a potential prototype for simulating Antarctic subglacial lake drainage.

Our model consists of a system of equations on a one-dimensional finite difference grid, with scalar values, such as ice-base

elevation, ice thickness, water pressure, and channel width calculated at the center of the grid cells, while fluxes of sediment and water are calculated at intermediate points. The systems of equations for the model are described in Sect. 3.1–3.3. Method of solution and time integration are described in Sect. 3.4. We describe our method of obtaining input data and the size of the domain in Sect. 3.5.

### 3.1 Theory for subglacial water flow

Subglacial water flows from areas of high hydropotential to areas of lower hydropotential. Hydropotential at the base of the ice, $\theta_0$ (calculated in meter water equivalent, m w.e.), is the sum of the water system elevation, $z_b$, and water pressure, $p_w$, normalized by water density, $\rho_w$, and gravity, $g$ (Shreve, 1972):

$$\theta_0 = z_b + \frac{p_w}{\rho_w g}. \tag{1}$$

Many models (e.g. Le Brocq et al., 2009; Carter et al., 2011) calculate $\theta_0$ assuming water pressure at the ice base is equal

to the overburden pressure, $p_o$, as quantities used to calculate $p_o$ (ice surface elevation $z_s$, $z_b$, and ice density $\rho_i$) are easily measured:

$$p_w = p_o = (z_s - z_b)\rho_i g. \tag{1a}$$

In reality, $p_w$ is the difference between the overburden pressure and effective pressure, $N$, which can only be calculated with seismic data (e.g. Blankenship et al., 1987) or borehole data (e.g. Engelhardt, 2004). Although $N$ does not typically affect

regional water routing, modelling work has shown that temporal change in $N$ is a critical part of the lake drainage process (e.g. Fowler, 1999; Evatt et al., 2006; Fowler, 2009). As $N$ decreases, it enables the formation of a temporary hydraulic divide between the lake and points downstream. As $N$ increases, water can then overcome the divide leading to the onset (and ongoing) drainage of that lake. Most of the theoretical work surrounding subglacial floods expresses hydropotential, $\theta$, in terms of pressure units (Pa), which in this work is calculated as follows:

$$\theta = z_b \rho_w g + (z_s - z_b)\rho_i g - N. \tag{1b}$$

While we also use $\theta$ (in Pa), we mostly express hydropotential as $\theta_0$ (in m w.e.) to allow more direct comparison between it and the measurements used to calculate it.



Given that $\theta$ and $N$, as well as channel cross-sectional area $(S)$, and water flux $(Q)$ are each defined differently for the R-channel, canal and distributed-flow systems, we will use the subscripts $_{RC}$, $_{CC}$, and $_S$, respectively, to avoid confusion in the following sections. Explanations for all symbols not defined explicitly in the text can be found in Tables 1 and 2.

## 3.2 Subglacial channel formation

5 The principle governing equations for the two channelization systems in our model all come from previous work: (i) the evolution of the R-channel, from Röthlisberger (1972) and Nye (1976); and (ii) the evolution of the canal model, from Walder and Fowler (1994) and Ng (2000). The method for solving both models has been largely adopted from recent work by Kingslake and Ng (2013), which specifically described the drainage of an ice-dammed lake via an R-channel coupled with a distributed system and thus was already well-posed for subglacial lake drainage. Exchange between channelized and distributed water systems in both models also follows Kingslake and Ng (2013) and is mediated by a term describing water transfer between the systems ($T_{RC}$ for the R-channel model; $T_{CC}$ for the canal model), which is a function of the pressure difference between the channelized and distributed systems. To fully adapt the Kingslake and Ng (2013) model to a domain containing Antarctic subglacial lakes, however, we made additional modifications, including defining the hydropotential gradient with Eq. (1), so that it is a function of the slope of the ice base and the effective pressure as well as the slope of the ice surface as defined in Kingslake and Ng (2013). We also allow for the change in pressure melting point of water with change in pressure (e.g. Alley et al., 1998; Hooke, 2005), as this effect has been reported to be significant to hydrological evolution over other flow paths following adverse bedslopes in Antarctica (e.g. Carter et al., 2009; Fricker et al., 2014).

### 3.2.1 Channels incised into the ice (R-channels)

In the classic R-channel model, transmissivity through a channel is controlled by the channel's cross-sectional area $(S_{RC})$, which is a balance of channel-wall melt-rate $(m_{RC})$ and viscous ice deformation rate $(C_{VRC})$ such that

$$\frac{\partial S_{RC}}{\partial t} = \frac{m_{RC}}{\rho_i} - C_{VRC}. \tag{2}$$

This formulation, originally presented in Röthlisberger (1972), has since been adapted for a variety of situations. The melt rate is related to the flux of water through the channel $(Q_{RC})$ by:

$$m_{RC} = Q_{RC} \frac{(1-k_h)\frac{\partial \theta_{RC}}{\partial x} + k_h \rho_w g \frac{\partial z_b}{\partial x} + k_h \frac{\partial N_{RC}}{\partial x}}{L_h}, \tag{3}$$

25 which describes the conversion of turbulent heat dissipation into melting with the term $k_h$ from Hooke (2005) to account for the change in melting point with pressure. The rate of viscous ice deformation into the channel is defined by:

$$C_{VRC} = K_{RC} S_{RC} N_{RC}^n, \tag{4}$$

which describes creep closure as a function of effective pressure $(N_{RC})$ and aperture size $(S_{RC})$, where $K_{RC}$ and $n$ are constants of Glen's flow law. Conservation of mass governs the change in flux along the flow path by:

$$\frac{\partial Q_{RC}}{\partial x} = m_{RC}\left(\frac{1}{\rho_w} - \frac{1}{\rho_i}\right) + C_{VRC} + T_{RC}, \tag{5}$$





while transfer between the R-channel and the distributed system (see Sect. 3.2.3) is governed by the transfer term $T_{RC}$ (Kingslake and Ng, 2013) using the equation:

$$T_{RC} = R_{kRC}k(N_{RC} - N_S), \tag{6}$$

where $R_{kRC}$ is the transfer efficiency, $k$ is a constant for connectivity (from Kingslake and Ng, 2013), and $N_{RC}$ and $N_S$ are the effective pressures of the R-channel and distributed system, respectively. Pressure along the channel co-evolves with water flux through an adaptation of the Manning friction formula, following Kingslake and Ng (2013):

$$\frac{\partial N_{RC}}{\partial x} = \rho_w g f_r \frac{Q_{RC}|Q_{RC}|}{S_{RC}^{8/3}} - \frac{\partial \theta_{RC}}{\partial x}, \tag{7}$$

where $f_r$ is the hydraulic roughness.

### 3.2.2 Channels incised into the sediment (canals)

Our model for channelization by mechanical erosion into the sediments is adapted extensively from concepts in existing models of lake drainage via R-channels eroded into the ice (e.g. Röthlisberger, 1972; Nye, 1976; Fowler, 1999; Evatt et al., 2006; Kingslake and Ng, 2013). It includes the basic principle of semi-circular channel capable of sustaining water pressures several m w.e. below flotation (Röthlisberger, 1972; Fowler, 1999), as well as more recent concepts regarding exchange of water between the channelized and distributed systems and evolution of pressure (Kingslake and Ng, 2013). In contrast to R-channel models, our model replaces melting of ice with sediment erosion and ice closure with deformation of sediment with linear viscous rheology, borrowing concepts and formulations from Walder and Fowler (1994) and Ng (2000). In describing channels eroded into the sediment, our work follows other efforts to model canals incised into the sediment behaving as conduits (e.g. van der Wel et al., 2013; Kyrke-Smith and Fowler, 2014) but, in contrast to these models, focuses specifically on subglacial lake drainage. Our descriptions of sediment erosion and deformation and of channel geometry are all extensively simplified; we regard the model developed below as a proof of concept that can be further refined if it is able to reproduce lake volume change as inferred from satellite and GPS observations (e.g. Fricker and Scambos, 2009; Siegfried et al., 2014, 2016) with realistic parameter choices.

As with the R-channel model, transmissivity for a channelized system with canals is governed by the canal cross-sectional area ($S_{CC}$). In our canal model, we ignore melting ($m_{RC}$) and deformation ($C_{VRC}$) of the ice above and assume all change to $S_{CC}$ is due to erosion and deformation of the sediment. Change in aperture in this model therefore is a balance of net erosion (erosion, $E_{CC}$, minus deposition, $D_{CC}$) and sediment deformation ($C_{VCC}$, for which we assume a linear viscous rheology):

$$\frac{\partial S_{CC}}{\partial t} = (E_{CC} - D_{CC})y_{CC} - C_{VCC}. \tag{8}$$

Adopted from Walder and Fowler (1994), $E_{CC}$ is a function of the mean sediment settling velocity ($v_s$), the channel geometry ($\alpha_{CC}$), the stress exerted on the bed by the flowing water ($\tau_{CC}$), and the particle size ($d_{15}$):

$$E_{CC} = K_{T1}\frac{v_s}{\alpha_{CC}}\left(\frac{\max(\tau_{CC} - \tau_k, 0)}{gd_{15}(\rho_{CC} - \rho_w)}\right)^{3/2}, \tag{8a}$$





where $K_{T1}$ is a sediment erosion constant and "max" refers to the maximum of $\tau_{CC} - \tau_k$ and zero. $\alpha_{CC}$ is a dimensionless correction factor between 5,500 and 233,000 to account for the geometric differences between the semi-circular channel geometry implied by the formulation and the actual geometry which is likely more elliptical in nature (e.g. Ng, 2000; Winberry et al., 2009; van der Wel et al., 2013).

$D_{CC}$, which also follows from Walder and Fowler (1994), is defined similarly:

$$D_{CC} = K_{T2}\frac{v_s}{\alpha_{CC}}\bar{c}\sqrt{\frac{gd_{15}(\rho_{CC} - \rho_w)}{\tau_{CC}}}. \tag{8b}$$

where $K_{T2}$ is a sediment deposition constant and $\bar{c}$ is the concentration of sediment in the water column.

Equations (8a) and (8b) imply that erosion and deposition are both a function of the hydraulic shear stress ($\tau_{CC}$), which is a function of water velocity. Following Walder and Fowler (1994), we use the mean water velocity ($u_{CC}$) such that

$$\tau_{CC} = \left|\frac{1}{8}f_{CC}\rho_w u_{CC}^2\right|, \tag{8c}$$

where $f_{CC}$ is a constant describing roughness. In our model, erosion begins once $\tau_{CC}$ exceeds a critical value, $\tau_k$, which is a function of median grain size as described by:

$$\tau_k = 0.025d_{15}g(\rho_{CC} - \rho_w). \tag{8d}$$

In this description, erosion of the sediments occurs only when water velocity exceeds a certain value since $\tau_{CC}$ is only depen-
dent on $u_{CC}$.

Sediment settling velocity ($v_s$), important for both Eq. (8a) and Eq. (8b), is treated as a function of sediment particle size ($d_{15}$), the density contrast between sediment and water ($\rho_{CC} - \rho_w$), and water viscosity ($\mu_w$), by the classic derivation from Stokes' Law (Lamb, 1932):

$$v_s = d_{15}^2\frac{2(\rho_{CC} - \rho_w)g}{9\mu_w}. \tag{8e}$$

Given that the net deposition rate as defined in Eq. (8b) is sensitive to the concentration of sediment already present, a mechanism is required for keeping track of sediment concentration. To achieve this, we assume that the transport rate of sediment downstream comes into equilibrium instantaneously. Sediment concentration ($\bar{c}$) at the upstream-most cell where erosion take place is given by

$$\bar{c} = y_{CC}\Delta x\frac{\rho_{CC}}{\rho_w}\frac{E_{CC} - D_{CC}}{Q_{CC}}. \tag{8f}$$

For each cell downstream, $\bar{c}$ is calculated as the sum of sediment eroded within the cell locally and the concentration of sediment in the water within the next cell upstream ($\bar{c}_{\text{up}}$):

$$\bar{c} = \bar{c}_{\text{up}} + y_{CC}\Delta x\frac{\rho_{CC}}{\rho_w}\frac{E_{CC} - D_{CC}}{Q_{CC}}. \tag{8g}$$





Closure occurs via viscous sediment creep, $C_{VCC}$, though this may in reality be a convenient continuum representation of discrete sediment collapse events on the sides of the channel:

$$C_{VCC} = \frac{|N_{CC}|}{N_{CC}} \frac{A_{CC} S_{CC} \left(\frac{|N_{CC}|}{a}\right)^a}{2N_\infty^b},$$ (8h)

where $N_\infty$ is the regional sediment effective pressure (which is closely correlated with sediment strength), $N_{CC}$ is the effective pressure of the water within the canal, and $A_{CC}$, $a$, $b$ are flow-law constants from Walder and Fowler (1994). $A_{CC}$ is similar to $K_{RC}$ in Eq. (4). Although our formulation for Eq. (8a–8h) is largely adapted from work by Walder and Fowler (1994), we follow Ng (2000) in decoupling $N_{CC}$ from $N_\infty$. In this work, however, we depart from Ng (2000) and treat $N_\infty$ as a constant and explore this simplifying assumption in more detail in Sect. 4.3.

In this system of equations, channel growth via erosion is a function of water velocity (through the relationship between erosion and shear stress) and is sensitive to the sediment size and the channel geometry. At lower water velocity, channelization in the sediment will not initiate and sheet flow will dominate. Channel closure via deformation is a function of the effective pressure ($N_{CC}$) and sensitive to the chosen value for sediment effective pressure ($N_\infty$). Although our formulation of Eq. (8h) theoretically allows for viscous channel growth if water pressure exceeds overburden pressure, such pressures are not expected (see also Sect. 4.1.3).

Conservation of water mass is accomplished by

$$\frac{\partial Q_{CC}}{\partial x} = -C_{VCC} + T_{CC}.$$ (9)

$T_{CC}$ is determined by the effective pressure difference between the canal and the distributed sheet via

$$T_{CC} = R_{KCC} k (N_{CC} - N_S),$$ (10)

with $R_{KCC}$ behaving analogously to $R_{KRC}$ (Eq. (6)). Finally we define the propagation of effective pressure ($N_{CC}$) along flow using a version of the Manning friction formula adapted from Kingslake and Ng (2013):

$$\frac{\partial N_{CC}}{\partial x} = \rho_w g \left( f_{CC} \frac{Q_{CC}|Q_{CC}|}{S_{CC}^{8/3}} \right) - \left( \frac{\partial \theta_{CC}}{\partial x} \right),$$ (11)

where we have introduced the roughness parameter $f_{CC} = f_r = 0.07$ m$^{-2/3}$s$^2$ based on the assumption of a semicircular channel geometry following Kingslake and Ng (2013).

### 3.2.3 Distributed sheet flow

The distributed sheet flow system, which is a component of both the R-channel and canal model, is governed by three primary equations all modified from Kingslake and Ng (2013): two concern the conservation of mass; a third governs the evolution of $N_S$. Water flux through the sheet ($Q_S$) is function of the hydraulic gradient ($\partial \theta_S / \partial x$) and cross-sectional area ($S_S$):

$$Q_S = S_S \left( \frac{\pi R_1}{4K_{S1}} \right)^{2/3} \left( \frac{6.6 \rho_w g}{f_r} \right)^{1/2} \left( \frac{\partial \theta_S}{\partial x} \right)^{-1/2},$$ (12)





where

$$S_S = y_S h_S. \tag{12a}$$

Equations (12) and (12a) assume flow path width, $y_S$, is constant width depth, such that water flux increases linearly with water layer thickness, $h_S$. Evolution of $S_S$ is governed by the conservation of mass:

$$\frac{\partial S_S}{\partial t} = M_C - \frac{\partial Q_S}{\partial x} - T, \tag{13}$$

where $M_C$ is a source term for water flowing into the system from the sides and $T$ is the flux between the sheet and channelized system, equal to either $T_{CC}$ or $T_{RC}$ depending on whether the distributed flow system is coupled to the canal or R-channel model.

Equation (12) implies that transmissivity increases linearly with water storage, while Eq. (13) implies that local water storage will increase if outflow from a cell is less than inflow. If water pressure is equal to overburden pressure then the only way for a water layer to exit an enclosed basin in the hydropotential is to thicken until it overtops the lowest saddle. With the inclusion of the effective pressure term ($N_S$) in the equation for hydropotential (Eq. (1b)), distributed water layers have another way over small obstacles. In a distributed system, $N_S$ tends to increase as water thickness decreases, by the relation:

$$N_S = \left[ \frac{\pi R_1 c_s n^n}{4 K_{RC}} \left( \frac{\tau_b^p}{S_S} \right) \right]^{\frac{1}{n+q}}. \tag{14}$$

Our parameterization for $N_S$ is adapted from the Kingslake and Ng (2013) description of a linked cavity system, but substitutes till cavities as observed by Engelhardt (2004) for hard bed cavities. Through this formulation, thinner water layers (and therefore higher values of $N_S$) are maintained over hydropotential maxima, while thicker water layers (and therefore lower values of $N_S$) are maintained over hydropotential minima. This coupling allows for a monotonically decreasing pathway in $\theta_S$ in spite of undulations in overburden pressure, which would inhibit the flow of water downstream without taking variable $N_S$ into account.

### 3.3 Modelling the lake

For a lake, we assume that the hydropotential is the sum of the ice base elevation ($z_b$) and overburden pressure (derived from $h_i$) at the centre of the lake (Figure 3). While hydropotential over the lake can then rise and fall in response to filling and drainage, we assume that the variation of hydropotential with time is spatially uniform across the lake. With the subscript $L$ referring to conditions (hydropotential, elevation change, etc.) across the lake, the change in lake level with time is defined as:

$$\frac{dz_{bL}}{dt} = \frac{dz_{sL}}{dt}. \tag{15}$$

Change in lake volume is calculated as:

$$\frac{\partial V_L}{\partial t} = Q_{\text{in}} - Q_{\text{out}}, \tag{16}$$



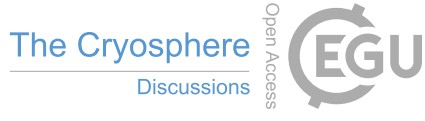

and

$$\frac{\partial \theta_L}{\partial t} = F_L \frac{Q_{\text{in}} - Q_{\text{out}}}{A_L}, \tag{17}$$

where $Q_{\text{in}}$ is inflow, $Q_{\text{out}}$ is outflow, $A_L$ is lake area (which is held constant in both the R- channel and canal models), and FL is a dimensionless number that varies between 1 and 2 that depending on the degree to which the overlying ice is supported by flexural bridging stresses (effectively a simplification of the parameterization of surface deformation in Evatt and Fowler (2007)). For large lakes that have a large section of freely-floating ice, where the zone of flexural deformation is less than 20% of the total lake area, FL = 1; Subglacial Lake Engelhardt (SLE) and Subglacial Lake Conway (SLC) fall into this category. For smaller lakes under thicker ice where the flexure zone comprises the most or all of the lake area and surface deformation can be approximated by a parabola, FL = 2; Subglacial Lake Mercer (SLM) and Subglacial Lake Whillans (SLW) fall into this category. We assume that there is no change in area as the lake state changes, and the implications of this assumption are explored in Sect. 4.2.2. We also assume that all surface displacement over the lake results directly from changes in lake volume, and neglect any temporal changes in ice thickness.

We also assume that there is only a single channel. Based on earlier runs of the R-channel and canal models as well as assertions in Shoemaker (2003), we assume that the destination lake is the nearest major low in the hydraulic potential, empirically defined to be an enclosed basin 3 m w.e. deeper than its surrounding. This assumption generally agrees with observations (e.g. Fricker et al., 2007). We neglect changes in the destination lake's level and assume that $N = 0$ at its centre. For the canal model, we also assume: (i) no change in sediment water content or rheology over time; (ii) negligible bridging stresses; (iii) near instantaneous advection of sediment downstream.

These systems of equations (2–17) are solved on a 1-dimensional finite difference domain, consisting of a source lake, intermediate points, and a destination lake. The point at which $\theta_0$ reaches a local maximum downstream of the lake is termed "the seal". This point is controlled by ice sheet geometry and is treated as immobile, in contrast to its definition in some higher-order models, which define the seal as the location where transmissivity is lowest at any given time (e.g. Fowler, 1999; Clarke, 2003).

### 3.4 Model implimentation

#### 3.4.1 Model spin up and initialization

Given our initial uncertainties about the onset of channelization, it is simpler to run our model when the source lake is filling, such that outflow from the lake is minimized, and the lake level is well below discharge-level. We initialize the model assuming that water pressure equals overburden pressure and that there is a constant supply of meltwater, $M_C$, along the flow path downstream of the lake, such that initial water flux in the distributed system ($Q_b$) increases with distance downstream of the lake. With these initial values, we calculate an initial water layer thickness, assuming $N_S = 0$, reworking Equation (12) such that:

$$S_S = Q_b \left( \frac{\pi R_1}{4K_{S1}} \right)^{-2/3} \left( \frac{6.6\rho_w g}{f_r} \right)^{-1/2} \left( \frac{\partial \theta_S}{\partial x} \right)^{1/2}. \tag{18}$$



We then use Eq. (14) to calculate a new value for $N_S$, based on $S_S$ and Eq. (13). For each subsequent time step we calculate $Q_S$ from Eq. (12), $\partial S_S / \partial t$ from Eq. (13), as well as change in lake properties from Eq. (15), (16), (17). With the resulting new water layer thickness, we finally recalculate $N_S$ and $S_S$.

### 3.4.2 Criteria for onset and shutdown of channelization

Initially water will flow from the seal towards the lake; as the lake level and lake hydropotential increase, however, water will begin flow out of the lake over the seal. Due to the inverse relationship between $N_S$ and $Q_S$, $Q_{out}$ will initially be much lower than $Q_{in}$. We assume that the initial outflow does not erode a channel at the seal but remains as sheet flow. Walder (1982) suggested that water velocity in a distributed system would be tend to be higher in places where irregularities in the overlying ice base and underlying bed lead to a thicker water layer. When the bed consists of sediments, $\tau_{CC}$ would exceed a critical threshold ($\tau_k$) in these locations before doing so elsewhere. As the water layer thickness increases erosion of the bed and/or melting of the ice above would be concentrated in these locations, leading to a positive feedback between local water layer thickness and water flow until a channel develops either in the bed below (a canal) or ice above (an R-channel).

Our model tries to simplify these complex dynamics by using a threshold criteria for channel initiation in which a channel carrying 0.5 m$^3$s$^{-1}$ appears spontaneously once $Q_S$ exceeds a critical threshold value ($Q_{\mathrm{onset}}$). Similar threshold methods have been employed in previous works, such as Flowers and Clarke (2002), Peters et al. (2009), de Fleurian et al. (2014), and Hoffman and Price (2014). The value for flux through newly formed channel is based empirically on work by Peters et al. (2009), which indicated that higher initial values would lead to an unrealistically rapid growth of outflow. As this value is also well below $Q_{onset}$, it allows for a gradual transition between a distributed and channelized system.

Effective pressure is initially set equal to the effective pressure in the distributed system so that $N_{RC} = N_S$ for the R-channel model (or $N_{CC} = N_S$ for the canal model). These initial values for water pressure then allow us to calculate an initial $S_{RC}$ (or $S_{CC}$) through Eq. (7) (or Eq. (11)). Cessation of channelization occurs once $Q_{RC}$ (or $Q_{CC}$) falls below a threshold value, $Q_{\mathrm{shutdown}}$, at which point it becomes computationally simpler to eliminate the channelization and revert back to sheet flow. Although it is possible that small incipient channels are always present such as is parameterized by Flowers et al. (2004), ignoring them improves computational speed. In Sect. 4.2, we explore in more detail how variations in $Q_{\mathrm{onset}}$ and $Q_{\mathrm{shutdown}}$ affect model performance.

### 3.4.3 Evolution of channelized flow

Once the channel is initiated we calculate the geometry of a proto-channel (Eq. (7), (11)), assuming $Q_{RC} = Q_{\mathrm{onset}}$ (or $Q_{CC} = Q_{\mathrm{onset}}$) everywhere between the source and destination lakes. After each successive iteration of Eq. (2), (3) and (4) (or Eq. (8), (9), and (10)), we recalculate $Q_{RC}$ and $N_{RC}$ (or $Q_{CC}$ and $N_{CC}$) along the flow path using a shooting method: beginning with an initial guess of $Q_{RC}$ (or $Q_{CC}$) at the outflow point, we use Eq. (7) (or Eq. (11)) to calculate $dN_{RC}/dx$ (or $dN_{CC}/dx$) locally on the staggered grid and the corresponding values for $N_{RC}$ and $T_{RC}$ (or $N_{CC}$ and $T_{CC}$) at the next point downstream on the regular grid. Using these newly calculated values for $N_{RC}$ and $T_{RC}$ (or $N_{CC}$ and $T_{CC}$), we calculate $dQ_{RC}/dx$ (or $dQ_{CC}/dx$) on the regular grid and then $Q_{RC}$ (or $Q_{CC}$) at the next point downstream on the staggered grid. This process is





repeated downstream until we calculate $N_{RC}$ (or $N_{CC}$) using Eq. (5) (or Eq. (9)) at the destination lake, at which point it is a known quantity. We compare our calculated $N_{RC}$ (or $N_{CC}$) with the "known" $N_{RC}$ (or $N_{CC}$) to obtain a misfit. We then use a Newton's method iteration on $N_{RC}$ (or $N_{CC}$) at the downstream lake (treated as a function of $Q_{RC}$ (or $Q_{CC}$) at the source lake outlet). After a maximum of 12 iterations we have arrived at a value for $Q_{RC}$ (or $Q_{CC}$) at the source lake that results in

a value for $N_{RC}$ (or $N_{CC}$) at the destination lake that is within 0.01 m w.e. of the known $N_{RC}$ (or $N_{CC}$). Once this value is obtained we then iterate Eq. (2), (6), and (7) (or Eq. (8), (10), and (11)) and the corresponding sheet flow evolution (Eq. (12), (13) and (14)). To improve model stability, we apply a modification of the Courrant-Friedrichs-Lewy (CFL) (Courant et al., 1967) criteria for the length of the time step. After calculating $dS_S/dt$ and $dS_{RC}/dt$ (or $dS_{CC}/dt$) we then define the time step $\Delta t$ by the relation:

$$\Delta t = \min \left( \left| \frac{S_S}{\frac{dS_S}{dt}} \right| 0.05, \left| \frac{S_{RC}}{\frac{dS_{RC}}{dt}} \right| 0.05, 10^5 \right), \tag{19a}$$

for the R-channel model, or

$$\Delta t = \min \left( \left| \frac{S_S}{\frac{dS_S}{dt}} \right| 0.05, \left| \frac{S_{CC}}{\frac{dS_{CC}}{dt}} \right| 0.05, 10^5 \right), \tag{19b}$$

for the canal model, where "min" refers to the minimum of all expressions within the parentheses.

### 3.5  Real and idealized model domains

Our idealized model domain (Figure 4a) was based on a simplified version of the flow path connecting the well-studied (e.g. Christianson et al., 2012; Horgan et al., 2012; Tulaczyk et al., 2014) Subglacial Lake Whillans (SLW) to the Ross Sea from Carter and Fricker (2012) (see Figure 1b for location; Figure 4b for comparison to the real domain). In this domain, the segment between the source lake and the seal, and the seal and the destination lake are approximated with straight lines. We tested both the R-channel and canal models on this domain and compared the model outputs. $y_S$ and $h_i$ were held constant at 2500 m and

500 m respectively. $A_L$ was set at 100 km$^2$. Based on water budgeting work by Carter et al. (2013), $Q_b$ was between 0.24 and 0.35 m$^3$s$^{-1}$. Values for other constants in this model run can be found in Table 2.

To demonstrate the ability of the canal model to reproduce the timing and magnitude of actual observed lake drainage events, we also applied it to several "real" domains for flow paths draining lakes in lower Whillans and Mercer ice streams, including SLW (Figure 4b), Lake Conway (SLC) and Lake Mercer (SLM) (Figure 4c) and Lake Engelhardt (SLE) (Figure 4d). For the

real-domains values, we obtained values for ice thickness, $h_i$, ice base elevation, $z_b$, and pathway width, $y_S$, from radar-derived measurements of ice thickness and surface elevation made between 1971 and 1999 over multiple campaigns (see Carter et al. (2013) for a description of the interpolation strategy and Lythe and Vaughan (2001) for descriptions of data used). For the idealized domain we used values of $\theta_0$, and $\theta_s$ at the source lake, seal and destination lake and then fit a straight line between them, holding ice thickness, and flow path width constant.





### 3.6 Summary of experiments

Our experiments began with several tests comparing the simulated lake volume time series output by the R-channel model against the volume change time series inferred from observations for SLW (Fricker and Scambos, 2009; Siegfried et al., 2014). After experimenting with the unaltered R-channel model we then experimented with perturbations to parameters controlling the rate of channel grown ($m_{RC}$) and closure ($C_{VRC}$) following hypotheses put forth in Fowler (2009) and Evatt et al. (2006) (Sect. 4.1.1).

All experiments involving lakes on a real domain were run until the the lake had undergone at least one complete fill-drain cycle and at least 10 years had elapsed. As the the sensitivity studies typically involved a small lake, they were run for only 8 years. In cases where the channel failed to grow the model was allowed to run for 10 years. In cases where runaway growth in outflow rate led to unrealistically large ice surface draw down, the model was stopped once the lake level had decreased by over 30 meters. In order to facilitate easier comparison of modeled lake volume change with that inferred from ICESat observations (which spanned 2003 - 2009) we referred to the time of observed volume change in years since 2000 and adjusted the timing of the first model year to coincide with the first observed filling cycle on the lake being simulated.

Following the suggestion that the channel closure rate inferred from lake volume change observations would be more consistent with the rheology of soft sediment, we focused subsequent experimentation on the canal model. Lake volume change as simulated by the canal model on a realistic domain was compared against lake volume change as observed for lakes SLW, SLE, SLC and SLM (Sect. 4.1.2). In order to explore the possibility of implementing the lake drainage model in areas where ice thickness measurements are more sparse, we compared the model's output on idealized (straight lines) and realistic (interpolated at 1 km intervals from RES and satellite altimetry) flow-path geometries (Sect. 4.1.3). The simplifications inherent in the idealized domain also provided useful for better quantifying how variations in the flow path geometries between the source and destination lakes might affected the magnitude and recurrence interval of lake drainage (Sect. 4.2.1). This idealized domain was also employed to explore how the time series output by the canal model varies in response to perturbations to both the input data, including inflow and lake area (Sect. 4.2.2), as well as changes to parameters that are more difficult to directly measure, such as $\alpha_T$, $d_{15}$, $N_\infty$, $R_1$, and $Q_{\text{onset}}$ (Sect. 4.2.3 and 4.2.4).

## 4 Results

### 4.1 Overview

All time series for lake volume change output by the models tested in this work were qualitative similarity to observed time series. Most models also reproduced key elements of the lake drainage process including initiation of outflow before the lake level reaches the flotation height, and the beginning of refilling without complete drainage of the lake. We begin with the results of the traditional R-channel model (Sect. 4.1.1), and then describe the results of the canal model (Sect. 4.1.2). We compare results from the real and idealized model domains (Sect. 4.1.3) and finally explore the models sensitivity to changes in key input parameters on the idealized domain (Sect. 4.2).



### 4.1.1 R-channel model versus observed height anomalies

Our lake drainage model based on outflow through an R-channel on the realistic SLW domain (Figure 4b) with realistic parameter choices (Table 1, 2) produced a single, large drainage event over the 30 year model run (Figure 5a). The long fill-drain cycle resulted in part from the presence of an adverse bed gradient along segments of the flow path. In these segments all of the turbulent heat generated by water traveling down flow went to maintaining the water at the pressure melting point rather than melting a channel (Figure 5b, 5c). From the start of the model run, it took nearly 10 years for a significant channel to begin growing, by which time the stiffness of the ice was too large to halt the lake drainage once the lake drained back to its initial level (defined as 0 m). Only after draining for almost 10 years and losing almost 16 m of elevation from its high stand did $Q_{out}$ fall below $Q_{in}$ and lake volume began increasing.

This modeled fill-drain cycle does not agree with observations from height anomalies at SLW (e.g. Fricker and Scambos, 2009; Siegfried et al., 2016), which show ~4 year fill-drain cycle (dashed line in Figure 5a, 5d). In an effort to match the observed height-change time series, we adjusted the parameter choices that controlled channel opening and closure (Figure 5d). By lowering the value for the latent heat of fusion ($L_h$), we were able to increase the channel opening rate and produce a timing of high stand that better approximated observations. However, higher melt rates continued to exceed channel closure rates and outflow increased exponentially. We adjusted the rate of channel closure by increasing $K_{RC}$, thereby increasing the rate of ice deformation into the channel. While softening the ice alone led to channels that never opened, when $K_{RC}$ was increased by a factor of 40 and $L_h$ was reduced by a factor of 400 (black line in Figure 5d) we were able to achieve a recurrence interval and magnitude of volume change similar to previous observations. Given that these values were roughly consistent with those found by Fowler (2009) (which suggested closure could occur if $K_{RC}$ was raised by two orders of magnitude), our finding supports the concept that most of the lake drainage events inferred from currently available observations are unlikely to occur via an R-channel.

### 4.1.2 Canal model versus observed height anomalies

For the real domains based upon SLW, SLE, SLC and SLM, our canal model with realistic parameter choices (Tables 1, 2) produced a simulated time series of lake volume that appeared qualitatively similar to the observed lake volume time series for each of these lakes with respect to the magnitude and recurrence interval of lake fill-drain cycles (Figure 6). At low stand, distributed flow carried water toward the lake. As $\theta_0$ at the lake increased with increasing $z_{bL}$, the difference between it and $\theta_0$ at the seal decreased until the seal was overtopped. At this point, a monotonically decreasing path from the lake over the seal could be maintained, allowing distributed flow out of the lake as lake level rose (Figure 3).

As the level of the lake increased further, the value for $N_S$ required to maintain a continuous path downstream decreased and fluxes from the distributed system increased. Even after the onset of channelized flow (i.e. $Q_{CC} > 0.5$ m$^3$s$^{-1}$), $Q_S$ initially remained closely correlated with lake volume, peaking slightly before the lake reached high stand. As channelized flow ($Q_{CC}$) increased however, the channel soon evolved to be the more efficient outlet from the lake. With further growth in $Q_{CC}$, lake level began to decrease more rapidly. Declining lake levels led to decreasing outflow via the distributed system as declining $\theta_S$





upstream led to a reduction in water inflow and in water layer thickness at the seal. A decline in $\theta_{CC}$ also caused a reduction in $Q_S$, allowing the channel then draw water away from what remained in the distributed system (Figure 6).

With further lowering of the lake level however, the gradient of $\theta_{CC}$ decreased, limiting the erosive power of the channel. Simultaneously, the increase in $N_{CC}$ allowed for more sediment to creep into the channel, decreasing channel size. While in

many model instances $Q_{CC}$ never ceased completely, it did eventually fall below the inflow level of the lake such that the lake could fill again. If a channel remained continuously open then a seal never developed in $\theta_{CC}$, only in $\theta_S$. With successive filling and draining cycles, growth in $S_{CC}$ lagged behind the level of the lake such that fluctuations in lake volume continued. In model runs where the channel remained open, the amplitude of volume change dampened through time (Figure 6d), a phenomenon explored in more detail in our sensitivity studies (Sect. 4.2.2).

We achieved a reasonable agreement between modelled and observed recurrence intervals and magnitude of lake volume changes for SLW, SLC and SLE using a constant values for $Q_{\text{in}}$ at each lake (Table 2; Figures 6a–d). However, our model could not reproduce observed periods of slower filling, such as was observed at SLE between 2006 and 2008 (Figure 6b). The weakest fit between model and observations occurred for SLM, for which our modeled frequency of filling and drainage events was over 50% higher than observed and ultimately out of phase. Work by Carter et al. (2013) has inferred that the filling rate

for SLM varied between 2.25 and over 50 m$^3$ s$^{-1}$ and was controlled primarily by outflow from SLC, suggesting that the misfit could reflect the poor assumption of non-varying $Q_{\text{in}}$. Achieving a simulated time series where the recurrence interval and magnitude of volume change were close those observed required that we vary the input parameters substantially from lake to lake (Table 2). We describe these sensitivities in more detail in Sect. 4.2

### 4.1.3   Canal model on the idealized versus realistic domain

A comparison between the model output from an idealized domain and one that uses more realistic flow path geometry showed almost no qualitative difference, with recurrence interval, fluxes and volume ranges within 5% of one another (Figure 7a, 7b). The primary difference between the two models was the length of time they took to run. While the linearly varying hydropotential of the idealized domain (Figure 7c) resulted in a nearly uniform cross-sectional area of the distributed system (Figure 7e), undulations in $\theta_0$ between the seal and the destination lake in the realistic domain (Figure 7d) resulted in large along-flow

variations in distributed system aperture (Figure 7f) and corresponding effective pressure. In particular, the advection scheme employed resulted in minor oscillations in the amount of water stored along the flow path, with more water stored in areas where $\theta_0$ was concave upward and less water stored in areas where $\theta_0$ was concave downward. On these local maxima, small changes to water storage still comprised a large portion the total and require a very short adaptive time step (see Eq. (17b)).

In both domains, values for $\theta_{CC}$ quickly converged upon a nearly constant gradient in $\theta_{CC}$ between the seal to the destination

lake (Figure 7c, 7d), and thus rates of water flux, channel growth and sediment deformation back into the channel were similar. As the canal transferred over 90% of the outflow downstream, the evolution of this feature was the most significant component for determining the timing and magnitude of lake volume change.





## 4.2 Model sensitivity

### 4.2.1 Sensitivity to flow-path geometry

Given that our lake drainage model is a bottleneck-style model in which the seal is stationary while the lake level rises in falls, we expect the rate of channel growth to correlate with hydropotential gradient, as well as the seal-height that must be overcome
initially. We tested the model's sensitivity to flow-path geometry by increasing (Figures 8a-c) and decreasing (Figures 8d-f) hydropotential gradients. We also explored both changes in total slope (Figures 8a, 8d) as well as the slope between the source lake and the seal (Figures 8b, 8e) and the seal and the destination lake (Figures 8c, 8f). Overall we found that steeper slopes favoured a more rapid, higher volume lake drainage with the slope downstream of the seal being the most important factor. Changes to the flow-path geometry upstream of the seal tended to be more important for determining the peak lake height
before drainage would occur (Figure 8b, 8e), with gentler slopes leading to higher high-stands and only a minor effect on the total volume range.

### 4.2.2 Sensitivity to inflow and lake area

At lower $Q_{in}$, there was an inverse linear relationship between recurrence cycle and input (Figures 9a, 9b). At higher $Q_{in}$, higher levels of inflow initially resulted in higher lake levels at high stand as the outflow channel took longer to grow large enough
to drain the higher inflow of water. While the higher high-stands and outflow levels initially provided a steeper hydropotential gradient resulting in a larger net volume loss, the higher inflow caused a net increase in volume when there was still a substantial quantity of water was leaving the lake. This process resulted in a lower net filling rate and lower high-stand volume during the next cycle, but a larger channel at low stand. As a consequence the magnitude of the fill-drain cycle dampened more rapidly with each successive cycle for higher values of $Q_{in}$. When we explored the effect of variations in $A_L$ (Figure 9c, 9d), we found
similar results with smaller lakes experiencing greater ranges in elevation initially, but then decreasing as outflow and inflow came into equilibrium.

### 4.2.3 Sensitivity to sediment and channel properties

Small changes to channel geometry, sediment effective pressure, and grain size ($\alpha_T$, $N_\infty$, $d_{15}$) significantly affected lake level at high stand, the magnitude of volume change, and the recurrence interval of drainage events (Figure 10). Moreover, the
variations in $N_\infty$ to which our model is sensitive (Figure 10b) are below the errors of current measurements (from borehole (e.g. Engelhardt, 2004) or seismic (e.g. Blankenship et al., 1987) experiments). Although we could find suitable values which resulted in a match between model runs and observation (Table 2), there is a significant element of non-uniqueness to these parameters, especially $\alpha_T$ and $d_{15}$, which are related through Eq. (8a) and (8b).



### 4.2.4 Lower sensitivity parameters

Any factor which affected the relationship between $h_S$ and $N_S$ (e.g. $d(z_s)/dx$, $R_1$, $y_S$) had an important influence on volume range as such factors control how easily water overcomes an apparent obstacle in the hydropotential. Any suite of parameter values that made it easy for water to clear apparent barriers had the effect of reducing the total amplitude of volume change, favouring steady drainage by sheet flow (Figure 11a). Conversely parameter values that made it difficult for water to overcome obstacles tended to increase the volume range and favour channelized drainage.

The range of values for $Q_\mathrm{onset}$ was limited by the maximum inflow (e.g. $\sim 5$ m$^3$s$^{-1}$ for SLW, inferred by Carter et al. (2013)). $Q_\mathrm{onset}$ had only minor effects when compared to the obstacle size ($R_1$) with respect to determining lake level at the onset of channelization and volume range. Given that differences between $Q_\mathrm{onset} = 1$ m$^3$s$^{-1}$ and $Q_\mathrm{onset} = 3$ m$^3$s$^{-1}$ were minimal, values for $Q_\mathrm{onset} < 0.5$ m$^3$s$^{-1}$ would likely be even less significant. Likewise, $M_c$ (inflow from other sources, such as basal melt and flow from outside the flow path) seemed to matter only when it was over an order of magnitude higher than values that would be reasonable for a location like the Whillans or Mercer ice streams. Under higher values of $M_C$, high stands were substantially lower but volume range was nearly unchanged (Figure 11b). When values for $M_C$ exceed 0.1 m$^3$s$^{-1}$ per km it affected the rate at which the source lake filled.

## 5 Discussion

Although our models suggest that a R-channel can theoretically grow and contract in the Antarctic subglacial environment supporting previous work by Carter et al. (2009) and Peters et al. (2009), their rates of growth and shutdown do no match those inferred for most active lakes unless the modeled melt and closure rates are adjusted significantly. If drainage through a canal system is the dominant mechanism for "active" lakes, then it would suggest that such lakes are more likely to be found in areas where soft sediment is widespread. In such an environment many of the criteria used to locate lakes with airborne radar might fail. Here we explore in more detail: (i) the validity and implications of the assumptions used in achieving our results; (ii) what might be inferred about Antarctic subglacial hydrology based on our results in the context of previous work; and (iii) the prospects for modelling subglacial lake drainage in ice sheet models.

### 5.1 Reassessment of model simplifications and assumptions

We did not include a model in which erosion and deformation of both the ice and bed occurred simultaneously. Given the difference in recurrence intervals predicted for Siple Coast style lake drainage events via the canal model versus an R-channel, we expect the evolution of the canal to dominate short-term flow variability. However, as the canal appeared to remain open throughout the fill-drain cycle, it is possible that ice deformation, melting, and freezing would all play a role in longer-term water-flow evolution. In particular if the basal ice is softer than would be implied by our assumed $K_{RC}$ of $10^{-24}$ Pa$^{-3}$s$^{-1}$ (Budd and Jacka, 1989) and/or the flow path follows a substantial adverse bedrock slope and is subject to high rates of basal freeze-on such as observed in Recovery Glacier (Fricker et al., 2014), then deformation of the ice into a canal could significantly



affect water flow and drainage system geometry. This process, which couples canals to ice deformation should be considered in future iterations of our model. Additionally, we have not explored the sensitivity of the model to the initial values for $Q_{RC}$ and $Q_{CC}$. Given that in most model runs the channel does not ever fall below $Q_{\text{shutdown}}$, it seems that alterations to this initial value for $Q_{RC}$ and $Q_{CC}$ would affect only the timing of the first filling and drainage cycle and thus any issues can be resolved with careful model spin-up.

One of the major challenges to our present model is the difficulty in measuring many of the properties to which our model is most sensitive. We have limited observations of grain size and sediment effective pressure observations (e.g. Tulaczyk et al., 1998; Engelhardt, 2004), limited channel geometry estimates from radar (e.g. Schroeder et al., 2013), and limited sediment effective pressure estimates from seismic surveys (e.g. Blankenship et al., 1987; Luthra et al., 2016). Even where present, estimates for these parameters are still not sufficiently accurate to effectively constrain our model. As a consequence, most of our simplifying assumptions were made due to data constraints, including those regarding the $A_L$ remaining constant over the filling-draining cycle, the till strength as a function of $N_\infty$ remaining constant over time, and our assumption of a semi-circular channel geometry for the canal. For example, $A_L$ likely correlates positively with lake volume. This effect alone would only moderately affect the rate of draw down leading to an increase in surface lowering relative to what would be expected in a constant $A_L$ case. Lower lake levels, however, would lead to faster rates of channel closure, partially counteracting this effect; these issues remain to be explored.

If $N_\infty$ is also increasing as the lake drains and shrinks, then we may expect the system to eventually come into equilibrium at low stand until flow from upstream increased. Other results exploring temporal changes in till strength (which indirectly related to $N_\infty$), however, suggest that $N_\infty$ is likely to be changing significantly over the duration of our model run as water is exchanged between the sediment pores and ice-bed interface (Christoffersen et al., 2014). When we consider our assumption that $N_\infty$ remains constant over time in light of these results, we can speculate on how variations in $N_\infty$ over time might affect the a subglacial lake's fill-drain cycle. Based on research investigating the exchange of water between the interfacial flow system and sediment (e.g. Christoffersen et al., 2014; Bougamont et al., 2015) we would expect $N_\infty$ to co-vary with $N_S$ and $N_{CC}$. In this situation, sediment strength would increase as lake level declined, $N_{CC}$ and $N_S$ increased, and water was removed from the surrounding sediment. As a result the channel might remain open longer than predicted by our model and low stands would last longer. A more detailed exploration of this process including in situ monitoring of sediment pressures may be key to improving the performance of our model.

The sensitivity of our model output to small variations in these parameters (as well as $\alpha_T$) calls into question our assumption that they remain constant with time. Ideally our model would fully incorporate a more complex array of glaciofluvial processes with spatial variation in sediment effective pressure and cohesion, all of which are related to till water content. Erosion, rather than being uniform along the channel, would then be concentrated in specific areas. Additionally, our experiments on R-channels suggest that melting, refreezing, and deformation of the overlying ice may change the channel significantly, especially in light of our results where the channels rarely cease once initiated. For this work however, the demonstration that reasonable rates for erosion and sediment deformation can produce changes in channel geometry sufficient to simulate volume change inferred from observations is a sufficient first step.



## 5.2 Comparison against other observations of lakes in Antarctica

Observations show that the high stand of active lakes is lower than the flotation heights of their dams (Christianson et al., 2012; Siegert et al., 2014). In all of our model runs, lake drainage initiated when $\theta_0$ was still well below $\theta_0$ at the seal (Figure 3). Our model is the first to capture this important dynamic for Antarctic subglacial lakes. We note that onset of outflow before

lake levels reach the flotation height has been previously observed and modeled for ice marginal lakes in alpine glaciers (e.g. Fowler, 1999) and the Grímsvötn caldera lake beneath the Vatnajökull ice cap in Iceland (Björnsson, 2003; Evatt and Fowler, 2007).

Our results are also consistent with a hypothesis put forth in Fowler (2009) that the change in lake level (and thus pressure) may be better explained if channels were incised into deformable sediments rather than ice. This mode of channelization has

two major implications: (i) the mechanism that allows active lakes to be inferred from satellite observations might make them difficult to detect with more traditional radar technology; and conversely (ii) lakes detected by more traditional radar sounding technology might drain via a different mechanism. If the rates of volume change inferred from satellite observations are better accommodated by channels incised in sediment, then it is likely that, in addition to the outlet of such a lake existing in soft sediments, the lake itself exists in soft sediments. The type of sediment that would both erode easily and quickly deform closed

would also tend to have a low angle of repose such that any lake formed within them would be shallow. This physical setting may explain why these lakes are not detectable using radar sounding (Gorman and Siegert, 1999), as a shallow lake in the presence of saturated sediment would fail the specularity test (Carter et al., 2007). Additionally, a lake surrounded by saturated sediments might not be significantly brighter than its surroundings as the reflection coefficient between ice and for saturated sediments is very close to that that of ice and lake water (Schroeder et al., 2013).

Our model for the R-channel suggests that such a mechanism dominates Antarctica subglacial lake drainage only in very specific circumstances. While Evatt et al. (2006) has explore the possibility of Antarctic subglacial lake drainage via such a mechanism, the recurrence interval they found (1000s of years between events versus a few decades of observations) would make it possible that we have yet to observe a lake draining via an R-channel. However, the drainage of Lake Cook E2 (Smith et al., 2009; McMillan et al., 2013; Flament et al., 2014), which resulted in a surface drawdown of over 50 m (similar in vertical

scale to observed subglacial lake drainages in Iceland), would be a promising candidate for drainage via an R-channel. Further study of lake drainage events outside of the Siple Coast is necessary to fully understand the respective roles of channels carved into ice and sediment.

## 5.3 Lakes in an ice sheet model

A number of recent models for ice flow have started to predict the formation of subglacial lakes in local hydropotential minima

(e.g. Sergienko and Hulbe, 2011; Goeller et al., 2013; Livingstone et al., 2013; Fried et al., 2014). These models all assume that these lakes simply fill until the water level reaches the flotation height of the "static seal" at which point they drain steadily through a distributed network at the ice bed interface. The results of both our R-channel and canal models indicate that channelized drainage is likely important for many of these lake systems. In particular, the fluctuations in volumes inferred



for lakes in fast flowing regions of the ice sheet are likely the result from channelization. As a consequence, ice flow in areas where subglacial lakes are common may have very different flow properties relative to a region where subglacial water flows in steady state (Stearns et al., 2008; Siegfried et al., 2016).

Regions of the ice sheet underlain by active subglacial lakes will likely exhibit more variability in ice flow rate with peak ice
velocity coinciding with peak distributed flow and peak lake volume (Figure 6). In chains of lakes, the lubrication may also be related to the evolution of the region hydrological system, rather than an individual lake (e.g. Siegfried et al., 2016). The exact degree to which lake drainage accelerates ice flow is, however, highly dependent on longitudinal stresses, which is outside the scope of this paper.

Spatial variations in basal traction in areas of fast flow have previously been proposed as a mechanism for forming lakes
(Sergienko and Hulbe, 2011), with lakes forming in the lee of local traction highs. More recently inversions of basal traction have inferred bands of stiff sediment impounding water in areas of moderate to fast flow, but usually outside the regions where active lakes are found (Sergienko et al., 2014; Hulbe et al., 2016). It may be that these categories represent two different mechanisms of water storage, one stable and the other unstable, resulting from subtle changes in sediment properties. Along with findings of multiple other mechanisms of water storage beneath the Antarctic ice sheet (Ashmore and Bingham, 2014),
these conclusions coupled with our results indicate that simply modelling the filling of an enclosed hydropotential depression, while an important first step, is not sufficient to simulate the full nature and impact of lake dynamics in an ice sheet model.

## 6   Summary

We have developed a new model for subglacial lake drainage beneath Antarctic ice streams in which channels are mechanically eroded into deformable subglacial sediment, and compared it to the lake volume change predicted by the more conventional
"R-channel" model, in which water incises into the base of the ice through melting. The "canal" model was better able to reproduce the timing and magnitude of subglacial lake drainage events in the well studied Whillans/Mercer ice stream system. Due to effective pressure differences between the lake centre and the "seal", lake drainage begins well before lake levels reaches flotation height and accelerates once flow is sufficient to initiate erosion into the sediment. Peak distributed flow correlated with lake level, while channelization is dominant when the rate of volume loss is highest. This evolution in the drainage system and
its impact on water pressures will likely influence local and regional ice dynamics.

The time series output by the model are highly sensitive to small changes in the properties of the subglacial sediments, in particular those related to sediment water content. Given recent work on the exchange of water between sediment pores and the interfacial flow system taking place over time frames of years to decades, it is likely that there are substantial changes in sediment strength over the filling or drainage cycle of a subglacial lake. It may even be that case that periodic filling and
drainage of subglacial lakes requires very specific sediment properties. Intriguingly recent analysis of sediment in lower WIS has not found substantial evidence for large scale fluvial deposition suggesting that the erosion necessary create subglacial channels occurrs only in limited places (Hodson2016)



The requirement of sediments for the realistic drainage of active lakes may explain why such lakes often fail classic radar detection criteria as lakes in such environments are often neither deep (and therefore not specular; Gorman and Siegert, 1999), nor significantly brighter than their surroundings (Carter et al., 2007). Conversely lakes detected by radar are typically found in steep-sided bedrock troughs (Wright and Siegert, 2012) and their surroundings likely lack the deformable sediments that

5 would facilitate episodic drainage on the timeframes for which observations exist. However, occasional drainage of such lakes via R-channels remains a possibility. For all lakes, the mechanisms for storage and release of water respond more to internal variability of the ice sheet than they do to external forcing. Consequently, the mechanism by which they drain and the effect of this mechanism on ice flow remains a critical uncertainty in predicting future ice sheet evolution.

*Acknowledgements.* Funding for this research was provided by the Antarctic Integrated Systems Science (AISS) program of the National

10 Science Foundation, through the WISSARD project (NSF grant ANT-0838885 (Fricker)) and by the NASA Cryospheric Sciences program.

Help working through the formulation benefitted from conversations with Geir Moholdt, and David Heezel. Conceptual development was improved by convserations with Richard Alley, and Robert Jacobel. Improvements to the formalization of the model benefitted from chats with Andrew Fowler, Mauro Werder, Tim Creyts, Christian Schoof, and Ian Hewitt. Additional context for this work was assisted by chats with Stephen Livingstone, Ted Scambos, Fernando Paolo, Adrian Borsa, Dustin Schroeder, Duncan Young, Slawek Tulaczyk, and WISSARD

15 Science Team members. Special thanks goes to Jonathan Kingslake who did an extensive review of a previous version of the manuscript with a patience and understanding possessed by only a few people on this planet. His efforts ultimately lead us to a much improved paper. Additional help with formatting the manuscript into LaTeX was provided by Anders Damsgaard.



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




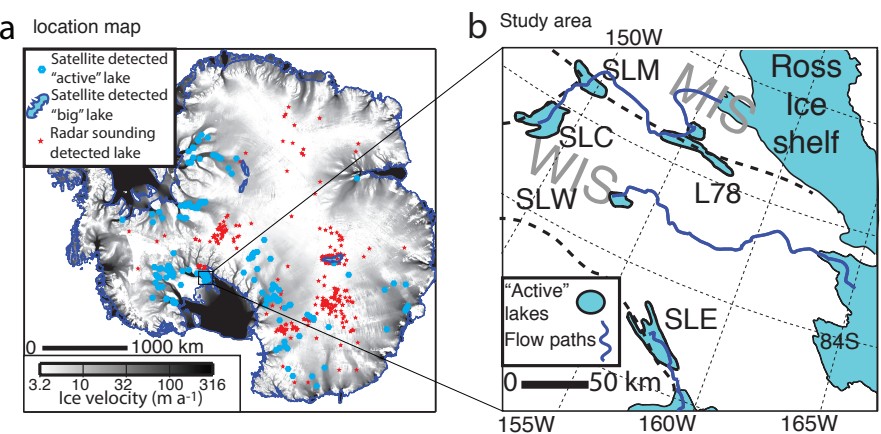

**Figure 1.** Maps of Antarctic subglacial lakes. (a) Antarctic subglacial lakes inventory showing detected "active" lakes, satellite detected "big" lakes, and radar sounding (RES) detected lakes, (Wright and Siegert, 2012). Background greyscale corresponds to ice velocities from Rignot et al. (2011). (b) Zoomed in map for Whillans/Mercer ice streams showing the lakes (SLM: Subglacial Lake Mercer, SLC: Subglacial Lake Conway, SLW: Subglacial Lake Whillans, SLE: Subglacial Lake Engelhardt, L78: Lake 7-8) and flow paths used in this study (see red box on panel (a) for location)



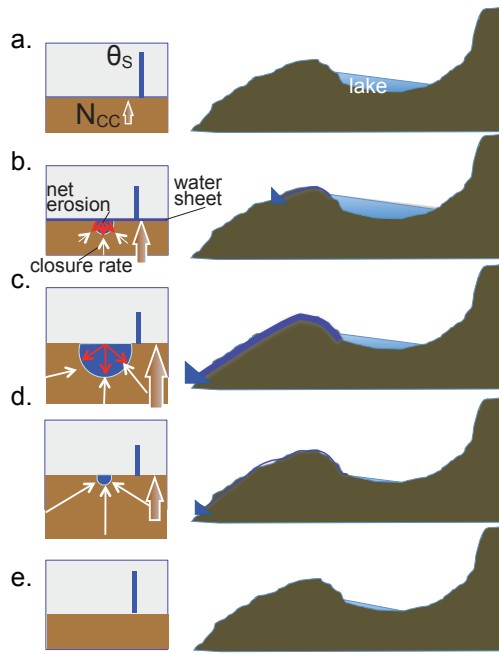

**Figure 2.** Schematic diagram showing the various stages of lake activity (right panels) and the corresponding evolution of the channel (left panels). Red arrow denotes net erosion rate, white arrow denotes closure rate, brown arrow denotes canal effective pressure, and blue bar indicates hydropotential in the distributed system: (a) initially the lake is filling and no outflow takes place; (b) as lake level increases, effective pressure differences between lake and seal allow for some water to escape via a distributed system (a.k.a. "water sheet"); (c) as the distributed system grows, a low pressure channel is eroded that begins to draw water form the surroundings; (d) while the low pressure channel allows for the lake to drain below the level at which outflow was initiated, as pressure lowers, sediment creeps inward; and (e) the channel contracts until its flow is negligible and the lake refills.





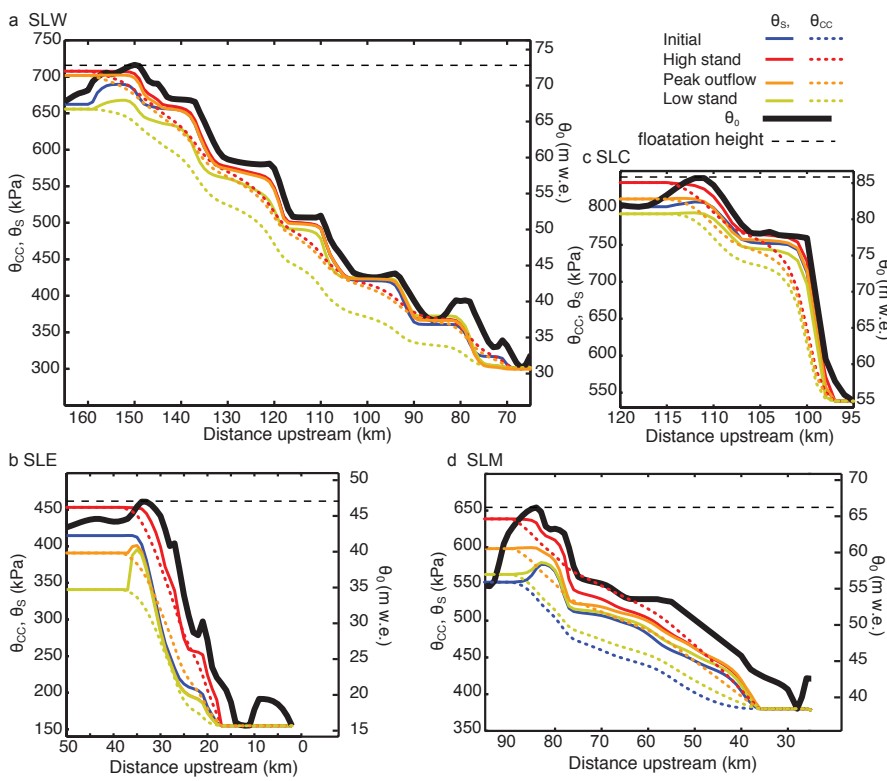

**Figure 3.** Along-flow hydropotential (left axis) in the distributed (solid) and channelized (dotted) drainage systems at four important time steps during a fill-drain cycle (initial state, high stand, peak outflow, low stand) for the four main lakes in the Whillans/Mercer subglacial system: (a) SLW; (b) SLE; (c) SLC; and (d) SLM. Thick black line (right axis) indicates overburden pressure calculated using Eq. (1a).





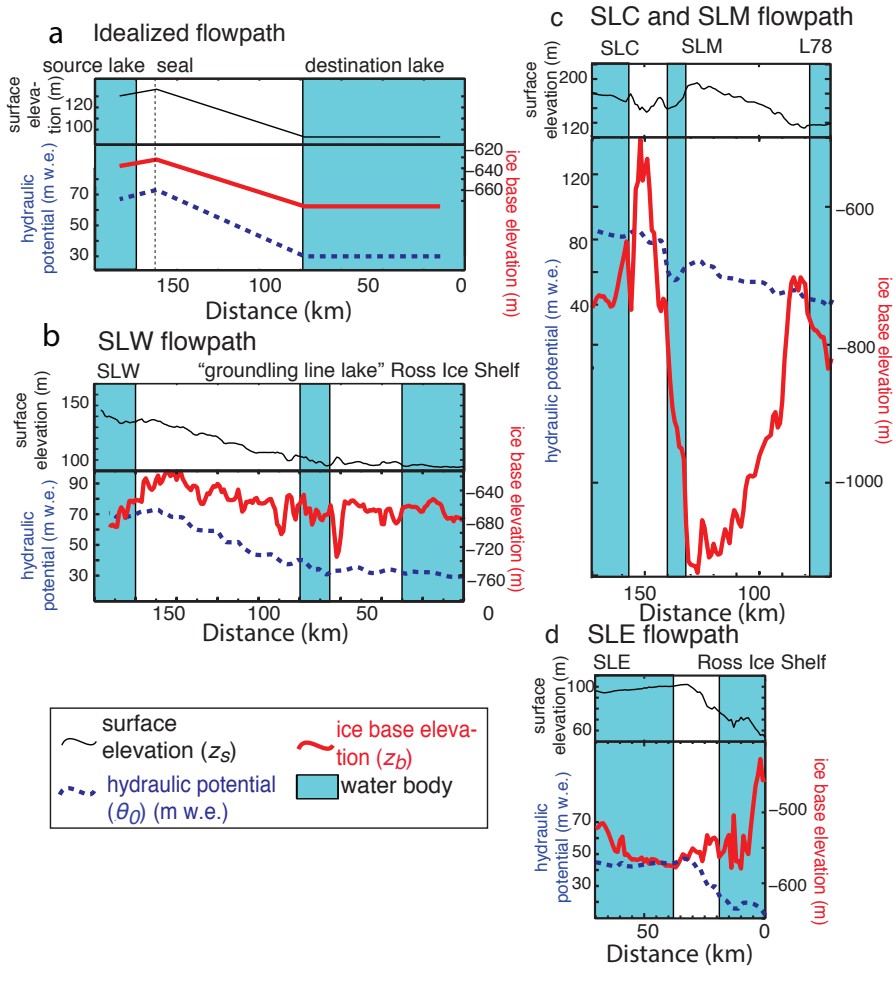

**Figure 4.** Ice surface elevation, hydropotential, and ice base elevation along the following flow paths (See Figures 1b for location of flow paths): (a) an idealized lake flow path based on SLW, (b) the flow path draining SLW (from Carter and Fricker, 2012), (c) the flow path draining SLC and SLM; and (d) the flow path draining SLE.

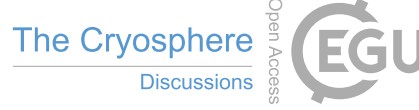

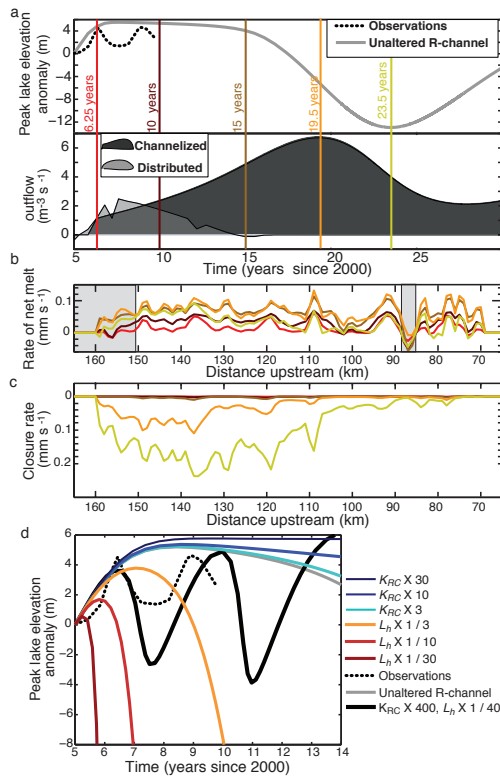

**Figure 5.** Results of experiments with R-channel model on the SLW domain (See Figure 1b for location and Figure 4b for flowpath geometry): (a) Modelled lake elevation anomaly and outflow obtained for the channelized and distributed drainage systems. Observed lake elevations (Carter et al., 2013) are also shown; (b) net melt rate along the flow path for five time steps shown in (a), with gray boxes denoting areas of net freezing; (c) viscous closure rate ($C_{VRC}$); and (d) modeled lake elevation anomalies resulting from varying parameters controlling channel closure ($K_{RC}$) and opening ($L_h$).

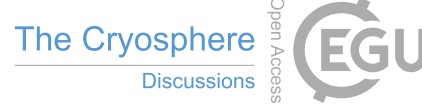



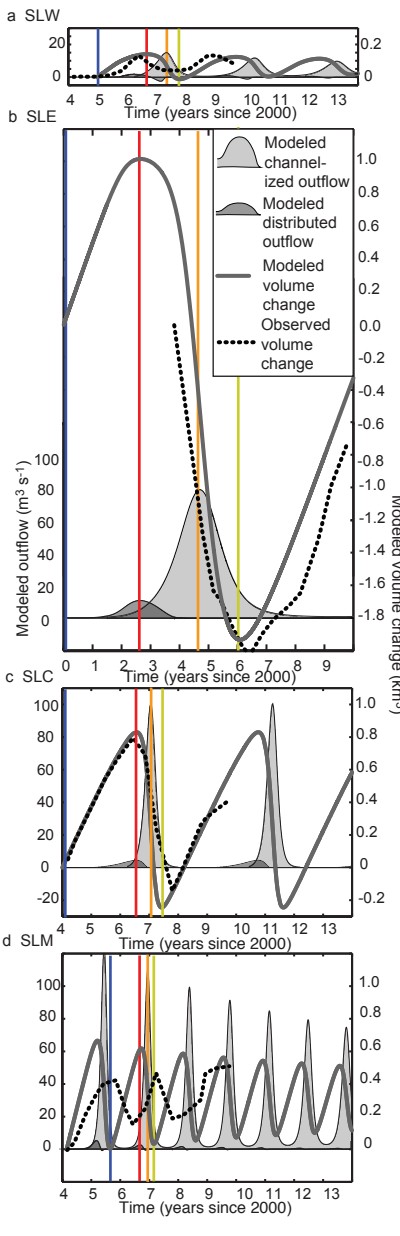

**Figure 6.** Results of experiments with canal model on the real domain (Figure 1b), showing evolution of modelled lake volume and outflow via distributed (grey) and channelized (blue) systems for constant $Q_{in}$ for the four main lakes: (a) SLW; (b) SLE; (c) SLC; and (d) SLM. Also shown are the observed lake volumes from (Carter et al., 2013). Coloured vertical lines correspond to same time steps from Figure 5.




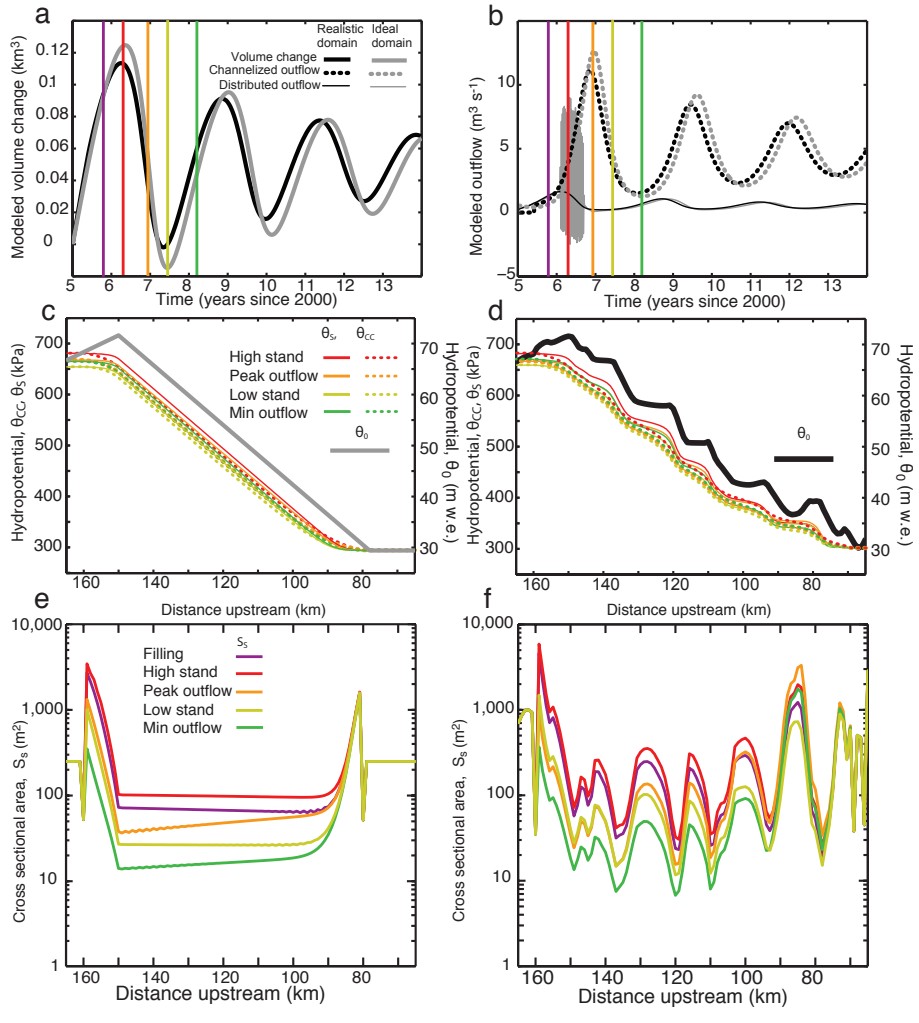

**Figure 7.** Comparison between canal model executed on the idealized and realistic domains. From top to bottom, pairs of panels show: (a) modelled volume change and (b) modelled outflow; evolution of hydropotential in the canal ($\theta_{CC}$) and in the distributed system ($\theta_S$) compared to ice-base hydropotential ($\theta_0$) on the (c) idealized and (d) realistic domains; and evolution of cross-sectional area of the distributed system ($S_S$) on the (e) idealized and (f) realistic domains.





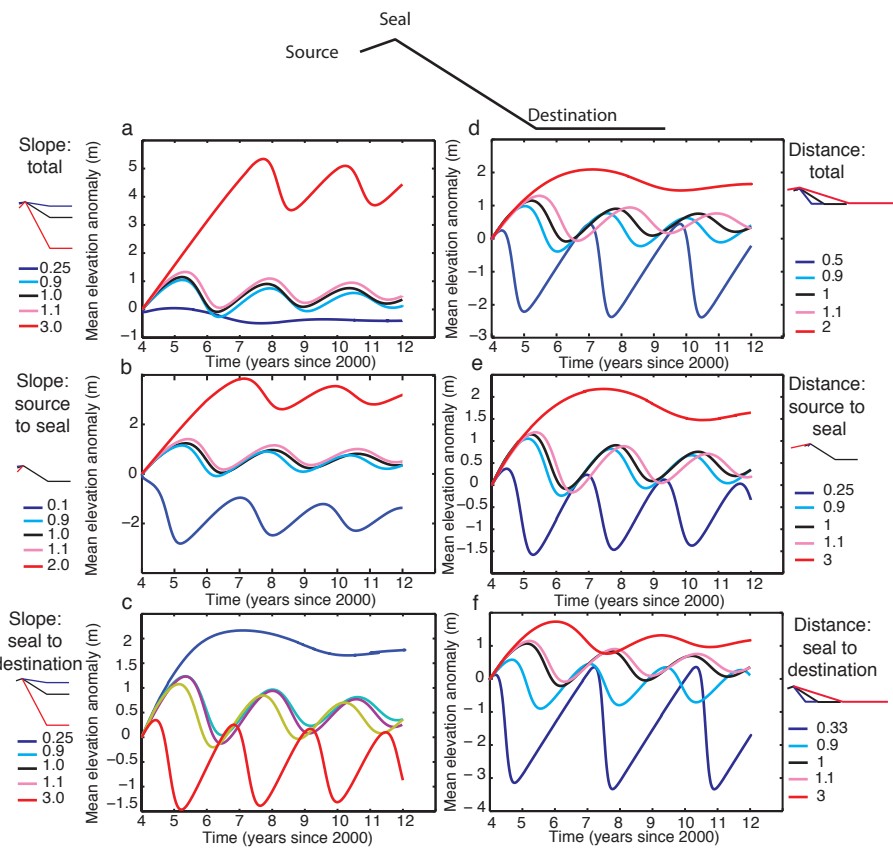

**Figure 8.** Sensitivity of canal model to flowpath geometry in the idealized domain. Left panels (a, b, c) show sensitivity of model output to changes in vertical scaling: (a) vertical exaggeration applied to the entire domain; (b) vertical exaggeration applied upstream of seal; and (c) vertical exaggeration applied downstream of seal. Right panels (d, e, f) show sensitivity of model output to changes in horizontal scaling: (d) horizontal stretching applied to the entire domain; (e) horizontal stretching applied upstream of seal; and (f) horizontal stretching applied downstream of seal.





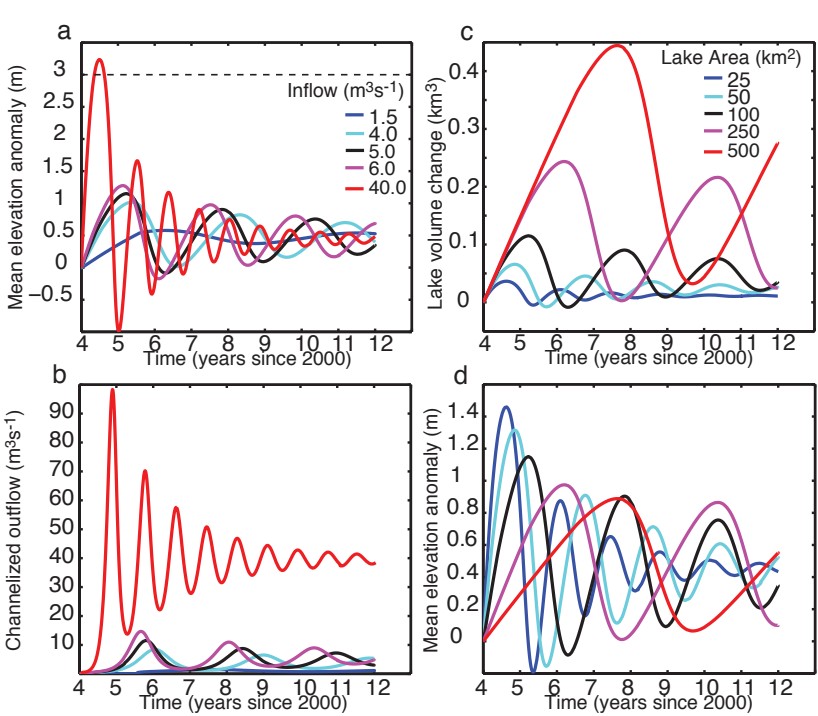

**Figure 9.** Sensitivity of canal model to inflow and lake area. Variation in (a) surface elevation anomaly ($z_{sL}$) and (b) channelized outflow ($Q_{CC}$) due to changes in inflow ($Q_{in}$). Variation in (c) lake volume ($V_L$) and (d) mean elevation anomaly ($z_{sL}$) due to changes in lake area ($A_L$). The black lines denotes the "control" run using our preferred parameter values (Table 2).





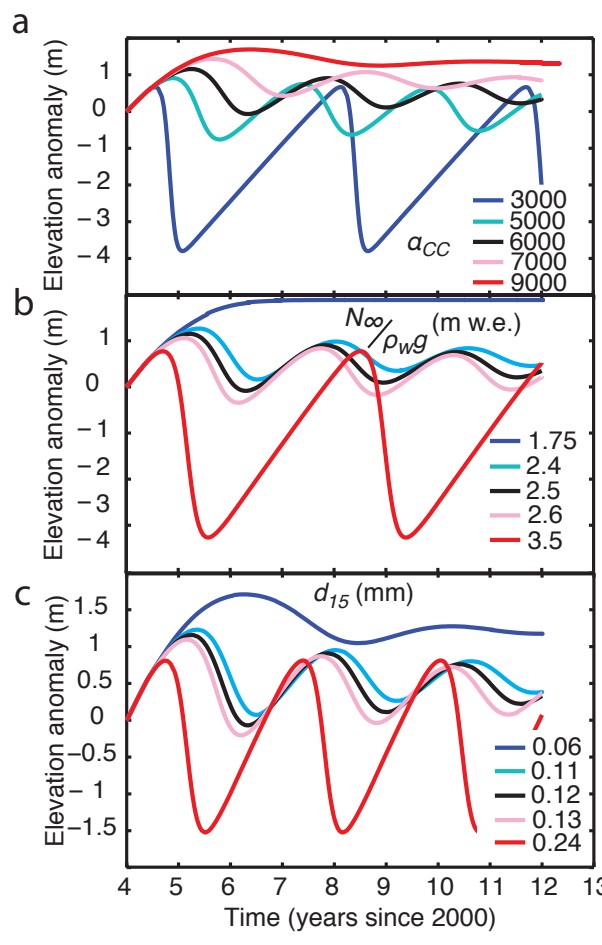

**Figure 10.** Sensitivity of canal model to sediment and channel properties. Variation of elevation anomaly ($z_{sL}$) due to five different values of (a) channel geometry ($\alpha_{CC}$), (b) sediment effective pressure ($N_\infty$), and (d) mean sediment grain size ($d_{15}$). The black lines denote the "control" run using our preferred parameter values (Table 2).





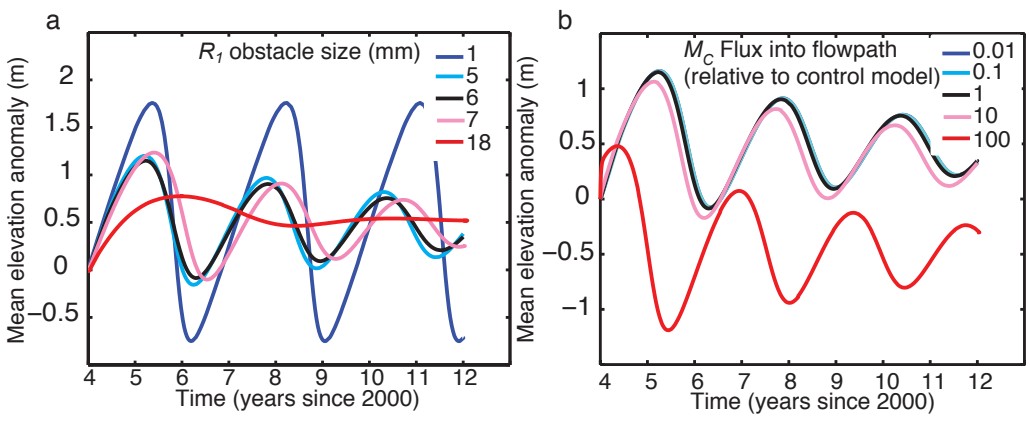

**Figure 11.** Sensitivity of canal model to (a) mean obstacle size ($R_1$) and (b) mean basal melt rate ($M_c$).



Table1: List of all symbols used in this paper with definitions, dimensions, and value/method used to estimate them.

| Symbol | Meaning | Dimensions | Value / method |
|---|---|---|---|
| $A_{CC}$ | Flow law constant for sediment | $M^{0.47}L^{-0.47}T^{-1.94}$ | $3 \cdot 10^{-5}$ Pa$^{b-a}$s$^{-1}$ |
| $A_L$ | Lake area | $L^2$ | Measured |
| $a$ | Constant of sediment deformation | - | 1.33 (Walder and Fowler, 1994) |
| $b$ | Constant of sediment deformation | - | 1.8 (Walder and Fowler, 1994) |
| $C_{VRC}$ | Rate of R-channel deformational closure | $L^2T^{-1}$ | Output by model |
| $C_{VCC}$ | Rate of viscous sediment closure | $L^2T^{-1}$ | Output by model |
| $\bar{c}$ | Sediment concentration in water column | - | Output by model |
| $c_S$ | Constant for roughness | $M^{-3}L^4T^5$ | $2 \cdot 10^{-20}$m s$^{-1}$ Pa$^{-3}$ |
| $D_{CC}$ | Deposition rate | $LT^{-1}$ | Output by model |
| $d_{15}$ | Characteristic grain size | L | Input (see Table 2) |
| $d_{50}$ | Median grain size | L | Calculated from input |
| $E_{CC}$ | Erosion rate | $LT^{-1}$ | Output by model |
| $f_r$ | Hydraulic roughness | $L^{-2/3}T^2$ | 0.07 m$^{-2/3}$s$^2$ |
| $f_{CC}$ | Canal roughness parameter | $L^{-2/3}T^2$ | 0.07 m$^{-2/3}$s$^2$ |
| $F_L$ | Coefficient of lake flexure | - | Input (between 1 and 2 based on observations reported in Fricker and Scambos, 2009) |
| $g$ | Gravitational acceleration | $LT^{-2}$ | 9.81 ms$^{-2}$ |
| $h_S$ | Water layer thickness for sheet flow system | L | Measured |
| $h_i$ | Ice thickness | L | Measured |
| $K_{RC}$ | Glen's flow law parameter for ice | $L^3T^5M^{-3}$ | $10^{-24}$Pa$^{-3}$ s$^{-1}$ (Kingslake and Ng, 2013) |
| $K_{S1}$ | Constant | - | 1.1 |
| $K_{T1}$ | Erosional constant | - | 0.1 |
| $K_{T2}$ | Constant of deposition | - | 6 |
| $k$ | Constant of sheet-conduit transfer | $M^{-1}L^3T$ | $10^{-9}$ m$^2$s$^{-1}$Pa$^{-1}$ (Kingslake and Ng, 2013) |
| $k_h$ | Coefficient for partitioning of turbulent energy between heating water and melting surrounding ice | - | 0.309 (Hooke, 2005) |
| $L_h$ | Latent heat of fusion | $L^2T^{-2}$ | 333,500 J kg$^{-1}$ |



| | | | |
|---|---|---|---|
| $M_c$ | Flux into system from melt/inflow | $L^2T^{-1}$ | Carter et al. (2013) |
| $m_{RC}$ | Röthlisberger channel melt rate | $ML^{-1}T^{-1}$ | Output by model |
| $m_{CC}$ | Canal net erosion rate | $ML^{-1}T^{-1}$ | Output by model |
| $N$ | Effective pressure | $ML^{-1}T^{-2}$ | Output by model |
| $N_S$ | Effective pressure in distributed system | $ML^{-1}T^{-2}$ | Output by model |
| $N_{RC}$ | Effective pressure in R-channel | $ML^{-1}T^{-2}$ | Output by model |
| $N_{CC}$ | Effective pressure in canal | $ML^{-1}T^{-2}$ | Output by model |
| $N_\infty$ | Effective pressure of sediments | $ML^{-1}T^{-2}$ | Input (see Table 2) |
| $n$ | Glen's flow law exponent | - | 3 |
| $p$ | Constant for power law sliding | - | 4 |
| $p_w$ | Water pressure | $ML^{-1}T^{-2}$ | Calculated |
| $Q_b$ | Initial flow in distributed system | $L^3T^{-1}$ | (Carter et al., 2013) |
| $Q_{in}$ | Inflow to lake | $L^3T^{-1}$ | (Carter et al., 2013) |
| $Q_{out}$ | Outflow to lake | $L^3T^{-1}$ | Output by model |
| $Q_S$ | Outflow via distributed system | $L^3T^{-1}$ | Output by model |
| $Q_{CC}$ | Outflow via canals | $L^3T^{-1}$ | Output by model |
| $Q_{RC}$ | Outflow via R-channels | $L^3T^{-1}$ | Output by model |
| $Q_{onset}$ | Outflow necessary for channel initiation | $L^3T^{-1}$ | Input (see Table 2) |
| $Q_{shutdown}$ | Outflow below which channelization ceases | $L^3T^{-1}$ | Input (see Table 2) |
| $q$ | Constant for power law sliding | - | 1 |
| $R_1$ | Characteristic obstacle height | $L$ | Input (see Table 2) |
| $R_{kRC}$ | Coefficient for transmission efficiency between R-channels and distributed flow systems | - | Input (0.05) |
| $R_{kCC}$ | Coefficient for transmission efficiency between canals and distributed flow systems | - | Input (0.05) |
| $S_S$ | Cross-sectional area of distributed system | $L^2$ | Output by model |
| $S_{RC}$ | Cross-sectional area of R-channel | $L^2$ | Output by model |
| $S_{CC}$ | Cross-sectional area of canal | $L^2$ | Output by model |
| $T_{RC}$ | Flux between distributed and R-channel systems per unit length | $L^2T^{-1}$ | Output by model |
| $T_{CC}$ | Flux between distributed and canal systems per unit length | $L^2T^{-1}$ | Output by model |
| $t$ | Time | T | Measured |




| | | | |
|---|---|---|---|
| $u_{CC}$ | Mean water velocity downstream | $LT^{-1}$ | output by model |
| $v_s$ | Mean settling velocity | $LT^{-1}$ | Calculated |
| $V_L$ | Subglacial lake volume | $L^3$ | Output by model |
| $x$ | Along flow distance | L | Measured |
| $y_S$ | Cross flow distance | L | Measured |
| $y_{CC}$ | Canal width | L | Calculated |
| $z_b$ | Ice base elevation | L | Measured initially, but changes with model output |
| $z_s$ | Ice surface elevation | L | Measured initially, but changes with model output |
| $z_{sL}$ | Ice surface elevation over the lake | L | Measured initially, but changes with model output |
| $z_{bL}$ | Ice base elevation over the lake | L | Measured initially, but changes with model output |
| $\alpha_{CC}$ | Geometry correction | - | Input (see Table 2) |
| $\theta$ | Hydropotential | $ML^{-1}T^{-2}$ | Measured/calculated |
| $\theta_0$ | Base hydropotential | L | Measured/calculated |
| $\theta_S$ | Hydropotential in distributed system | $ML^{-1}T^{-2}$ | Measured/calculated |
| $\theta_{RC}$ | Hydropotential in R-channel | $ML^{-1}T^{-2}$ | Measured/calculated |
| $\theta_{CC}$ | Hydropotential in canal | $ML^{-1}T^{-2}$ | Measured/calculated |
| $\theta_L$ | Hydropotential in lake | M | Measured/calculated |
| $\mu_w$ | Viscosity of water | $ML^{-1}T^{-1}$ | $1.787 \cdot 10^{-3}$ Pa s |
| $\rho_w$ | Density of water | $ML^{-3}$ | 1000 kg m$^3$ |
| $\rho_i$ | Density of ice | $ML^{-3}$ | 917 kg m$^3$ |
| $\rho_{CC}$ | Density of sediment | $ML^{-3}$ | 2700 kg m$^3$ |
| $\tau_b$ | Basal driving stress | $ML^{-1}T^{-2}$ | Calculated from initial measurements |
| $\tau_k$ | Critical hydraulic shear stress necessary to initiate erosion | $ML^{-1}T^{-2}$ | Calculated |
| $\tau_{CC}$ | Hydraulic shear stress | $ML^{-1}T^{-2}$ | Output by model |



Table 2: List of parameters used for each of the experiments.

| | Idealized/control | R-channel | SLW | SLE | SLC | SLM |
|---|---|---|---|---|---|---|
| $\alpha_T$ | $6.0 \cdot 10^3$ | N/A | $1.9 \cdot 10^4$ | $2.3 \cdot 10^5$ | $1.3 \cdot 10^4$ | $6.0 \cdot 10^3$ |
| $d_{15}$ [mm] | 0.12 | N/A | 0.24 | 1.5 | 0.25 | 0.12 |
| $Q_{\text{in}}$ [m$^3$s$^{-1}$] | 5 | 5 | 4 | 16 | 12 | 22.5 |
| $Q_{\text{onset}}$ [m$^3$s$^{-1}$] | 0.75 | 0.75 | 1.75 | 1.25 | 3.5 | 0.75 |
| $Q_{\text{shutdown}}$ [m$^3$s$^{-1}$] | 0.25 | 0.25 | 0.25 | 0.25 | 0.25 | 0.25 |
| $R_1$ [mm] | 6.0 | 6.0 | 6.0 | 24 | 1.5 | 6.0 |
| $A_L$ [km$^2$] | 100 | 100 | 58 | 257 | 247 | 132 |
| $N/\rho_w g$ [m w.e.] | 2.5 | N/A | 3.25 | 9 | 1.75 | 2.5 |
| $H_L$ | 2 | 2 | 2 | 1 | 1 | 2 |
| $M_C$ [m$^3$s$^{-1}$] | 0.001 | 0.001 | 0.001 | 0.013 | 0.025 | 0.002 |