# Peer review of "Antarctic subglacial lakes drain through sediment-floored canals: Theory and model testing on real and idealized domains"

_The Cryosphere, 2016_

## Referee Comment (RC1) · J. Kingslake (Referee) · 22 Jun 2016

**Review of 'Antarctic subglacial lakes drain through sediment-floored canals: Theory and model testing on real and idealized domains' by Sasha. P. Carter, Helen. A. Fricker and Matthew. R. Siegfried**

**Jonathan Kingslake**

I reviewed a previous version of this manuscript (doi:10.5194/tcd-9-2053-2015), and under the direction of the editor have restricted my comments below to the responses of the authors to my comments.

The authors have done an excellent job of responding to my comments and the revised paper is hugely better than the original submission.

I thank the authors for taking my comments so seriously and persevering through what must have been a hard revision process. I am really pleased that they did persevere and, as I say, in my opinion, the paper is much better for it. It now presents a more compelling argument for the drainage of active lakes being controlled by sediment erosion and deformation rather than ice melting and creep.

What follows is a discussion point that might be interesting and a few comments on the details of the rebuttal.
* * *
One exciting implication that occurred to me when reading an added sentence (page 17, lines 14-16: "Work by Carter et al. (2013) has inferred that the filling rate for SLM varied between 2.25 and over 50 m3 s−1 and was controlled primarily by outflow from SLC, suggesting that the misfit could reflect the poor assumption of non-varying Qin."), is related to the predictability of lake drainage.

I think that one of the things that this work supports is the idea that the filling and drainage of subglacial lakes is controlled by the same fundamental physics as those described by more traditional models (e.g. Fowler, 1999; Evatt et al., 2006, Kingslake and Ng, 2013; Kingslake, 2015), which only really considered R-channels. In the 2015 paper (Kingslake, 2015; Kingslake, J., 2015. Chaotic dynamics of a glaciohydraulic model. Journal of Glaciology, 61(227), pp.493-502.) I showed that an R-channel model can behave in a few interesting ways like nonlinear oscillators when it is supplied by a time-varying input - i.e. it can be chaotic and fundamentally unpredictable beyond a certain time in the future. In that paper I speculated that this could happen in subglacial lakes, but I stopped short of speculating on the implication for the fundamental unpredictability of ice-sheet dynamics.

Carter et al mention in the sentence quoted above ("...filling rate for SLM varied between 2.25 and over 50 m3 s−1 and was controlled primarily by outflow from SLC...") that lake input is controlled by outflow from other lakes. This is exactly what I speculated would be needed for chaotic dynamics to be produced by a subglacial lake system. Now that it has been shown that the same fundamental physics apply to subglacial lakes (albeit with effective-pressure-dependent viscous ice flow replaced by effective-pressure-dependent viscous sediment creep and discharge-dependent ice melt replaced by discharge-dependent sediment erosion), perhaps this connection is worth thinking about.

It potentially says something quite fundamental about the predictability of ice sheets! Because water pressures control ice flow and because lake drainage and filling controls water pressure and because lake drainage and filling could be chaotic, could ice-sheet dynamics behave chaotically? It would be

quite fun to hypothesize that the details of ice-sheet dynamics can never be predicted beyond a few fill-drain cycles into the future.

Anyway, just a suggestion, but maybe the authors would like to think about these ideas and maybe add a paragraph in the discussion if they think it's interesting.
* * *
From table 1, the parameter $R_{kRC}$ is equal to 0.05. Does this mean that the transfer between drainage systems is 20 times smaller than in previous work (Hewitt and Fowler, 2009; Kingslake and Ng, 2013)? Admittedly these values are highly uncertain, but I was thinking that this might be the explanation for the weak sensitivity to the distributed system supply term MC.

I think a typo remains in eqn 12 after the correction. Should the exponent of $(d\theta\_s/dx)$ be ½ rather than -1/2, so that discharge increases with hydraulic gradient?

A small point is that the subscript 'C' in the source term in the eqn (13) has not been changed as mentioned in the rebuttal.

It has not been explained that eqn 14 assumes steady-state.

I cannot find the following passage that is mentioned was added in the rebuttal: "If the model was allowed to continue to run for longer timeframes, however, then it was possible for discharge to increase. Even in a domain with a perfectly horizontal ice base the channel still grew too slowly taking 12 years to drain back to the initial lake level (Figure 5b, 5c)."

I am sorry to say that I still do not understand eqn 7. If you differentiate eqn 1b to get $d\theta/dx$ and substitute this into eqn 7, it seems to me that $dN/dx$ cancels and you are left with an equation that does not include the effective pressure.
* * *
In summary, I am really pleased that the authors have produced such an interesting and well-presented paper. I expect it will be well-read and useful and as I mentioned above it is interesting to think about its immediate implications for, among other things, the predictability of ice sheets.

Best wishes,

Jonny

---

## Editor Comment (EC1) · R.G. Bingham (Editor) · 12 Jul 2016

This paper – now doi:10.5194/tc-2016-74 – in fact represents a significant revision of a previous paper submitted to The Cryosphere Discussions, doi:10.5194/tcd-9-2053-2015.

As the Editor of the new submission, in addition to reading the paper I have reviewed (a) the Open Discussion on http://www.the-cryosphere-discuss.net/tc-2015-17/, which documents the three extremely detailed reviews and an additional comment that the authors received for version tcd-9-2053-2015; (b) the authors' rebuttal of the three original reviewers' comments submitted as part of the further review process of tcd-9-2053-2015; and (c) a further review of the revised text, also submitted as part of the

further review process of tcd-9-2053-2015.

From reading these documents together with the new paper (i.e. tc-2016-74), I am satisfied that this submission forms, effectively, a continuation of the review process that commenced with tcd-9-2053-2015, and which had already reached an advanced stage. Consequently, this submission has had need of only one review (a re-review from one of the reviewers of tcd-9-2053-2015) beyond my Editorial Review, rather than at least two as normally required for new submissions to The Cryosphere.
* * *

---

## Author Comment (AC1) · 2 Aug 2016

*Dear Dr. Bingham,*

*Thank you for the time, consideration and assistance throughout this process. Below is our response to the reviewer. Reviewer comments are in regular type, our responses are in italics*

Review of 'Antarctic subglacial lakes drain through sediment-floored canals: Theory and model testing on real and idealized domains' by Sasha. P. Carter, Helen. A. Fricker and Matthew. R. Siegfried

Jonathan Kingslake

I reviewed a previous version of this manuscript (doi:10.5194/tcd-9-2053-2015), and under the direction of the editor have restricted my comments below to the responses of the authors to my comments.

The authors have done an excellent job of responding to my comments and the revised paper is hugely better than the original submission. I thank the authors for taking my comments so seriously and persevering through what must have been a hard revision process. I am really pleased that they did persevere and, as I say, in my opinion, the paper is much better for it. It now presents a more compelling argument for the drainage of active lakes being controlled by sediment erosion and deformation rather than ice melting and creep. What follows is a discussion point that might be interesting and a few comments on the details of the rebuttal.

*We thank you for your time energy and consideration. This paper truly could not have been what it has become without your insight and attention that went far beyond the call of the standard reviewer. Reading your words has made all the efforts feel worthwhile and we hope more good does indeed come of it.*

One exciting implication that occurred to me when reading an added sentence (page 17, lines 14-16:

"Work by Carter et al. (2013) has inferred that the filling rate for SLM varied between 2.25 and over 50 m3 s−1 and was controlled primarily by outflow from SLC, suggesting that the misfit could reflect the poor assumption of non-varying Qin."), is related to the predictability of lake drainage. I think that one of the things that this work supports is the idea that the filling and drainage of subglacial lakes is controlled by the same fundamental physics as those described by more traditional models (e.g. Fowler, 1999; Evatt et al., 2006, Kingslake and Ng, 2013; Kingslake, 2015), which only really considered R-channels. In the 2015 paper (Kingslake, 2015; Kingslake, J., 2015. Chaotic dynamics of a glaciohydraulic model. Journal of Glaciology, 61(227), pp.493-502.) I showed that an R-channel model can behave in a few interesting ways like nonlinear oscillators when it is supplied by a time-varying input - i.e. it can be chaotic and fundamentally unpredictable beyond a certain time in the future. In that paper I

speculated that this could happen in subglacial lakes, but I stopped short of speculating on the implication for the fundamental unpredictability of ice-sheet dynamics.

1. Carter et al mention in the sentence quoted above ("…filling rate for SLM varied between 2.25 and overm50 m3 s−1 and was controlled primarily by outflow from SLC…") that lake input is controlled by outflow from other lakes. This is exactly what I speculated would be needed for chaotic dynamics to be produced by a subglacial lake system. Now that it has been shown that the same fundamental physics apply to subglacial lakes (albeit with effective-pressure-dependent viscous ice flow replaced by effective pressure-dependent viscous sediment creep and discharge-dependent ice melt replaced by discharge dependent sediment erosion), perhaps this connection is worth thinking about. It potentially says something quite fundamental about the predictability of ice sheets! Because water pressures control ice flow and because lake drainage and filling controls water pressure and because lake drainage and filling could be chaotic, could ice-sheet dynamics behave chaotically? It would be quite fun to hypothesize that the details of ice-sheet dynamics can never be predicted beyond a few fill drain cycles into the future.  Anyway, just a suggestion, but maybe the authors would like to think about these ideas and maybe add a paragraph in the discussion if they think it's interesting.

*In short, we have thought about this a lot, especially in light of your recent J. Glac paper on chaotic lake drainage dynamics.  Indeed, we actually are working on a model where inflow varies with time and, when several lakes are chained together, the dynamics can turn chaotic quite quickly. However, we felt this next step was beyond the scope of the current paper, which was establishing that the canal model as a viable alternative to R-channels, and therefore did not include it in this manuscript.*

2. From table 1, the parameter RkRC is equal to 0.05. Does this mean that the transfer between drainage systems is 20 times smaller than in previous work (Hewitt and Fowler, 2009; Kingslake and Ng, 2013)?  Admittedly these values are highly uncertain, but I was thinking that this might be the explanation for the weak sensitivity to the distributed system supply term MC.

*Our value for RkRC is indeed really small.  We struggled with model compilation at higher values.  We now include the following sentence in Methods (p. 8):*

*"It should be noted that our value for $R_{kRC}$ is near the lowest end of values explored by Kingslake and Ng (2013) primarily due to model stability issues."*

*And in Results (p. 19):*

*"This low sensitivity lmayikely results from our low value for $R_{kRC}$  which limited the transfer of water between the channelized and distributed systems"*

3. I think a typo remains in eqn 12 after the correction. Should the exponent of $(d\theta_s/dx)$ be ½ rather than -1/2, so that discharge increases with hydraulic gradient?

*We inspected the equation as it appears in the paper and in the code and agree with you that -1/2 should be 1/2. We also corrected the preceding term in this equation, where sqrt(6.6 rho_w g / f_r) was meant to be sqrt(6.6 / rho_w g f_r). The latter representations are all consistent with how this was coded. We apologize for these typos.*

4. A small point is that the subscript 'C' in the source term in the eqn (13) has not been changed as mentioned in the rebuttal.

*We have now changed the subscript to 'S' everywhere to maintain consistency with other variables related to the distributed system.*

5. It has not been explained that eqn 14 assumes steady-state.

*After eqn 14 we have now added the language, "This formulation, assumes $N_S$ reaches steady-state instantaneously. Thinner water layers (and therefore higher values of $N_S$) are maintained over hydropotential maxima, while thicker water layers (and therefore lower values of $N_S$) are maintained over hydropotential minima."*

6. I cannot find the following passage that is mentioned was added in the rebuttal: "If the model was allowed to continue to run for longer timeframes, however, then it was possible for discharge to increase. Even in a domain with a perfectly horizontal ice base the channel still grew too slowly taking 12 years to drain back to the initial lake level (Figure 5b, 5c)."

*This is likely an issue related to differences between the manuscript you reviewed (submitted in March 2016) and a previous version that responded to your original review (submitted in December 2015). Looking through the response letter, the original comment concerned discrepancies between the figure illustrating the R-channel dynamics and what was written in the text. The dynamics of the R-channel are now illustrated in Figure 5; the text quoted above was removed from the manuscript and was replaced by the following language concerning the time necessary for outflow to exceed inflow:*

*"From the start of the model run, it took nearly 10 years for a significant channel to begin growing, by which time the stiffness of the ice was too large to halt the lake drainage once the lake drained back to its initial level (defined as 0~m). Only after draining for almost 10 years and losing almost 16~m of elevation from its high stand did $Q_{out}$ fall below $Q_{in}$ and lake volume began increasing."*

7. I am sorry to say that I still do not understand eqn 7. If you differentiate eqn 1b to get $d\theta/dx$ and substitute this into eqn 7, it seems to me that $dN/dx$ cancels and you are left with an equation that does not include the effective pressure.

*We took a close look at this equation in the paper and in the code. We have now rewritten the last term as "rho_w g\frac{\partial \theta_{0}}{\partial x}"*

*With the reformulated equation, the hydropotential gradient in the channel (partial theta_RC / partial x) is equal to the base hydropotential gradient (rho_w * g partial theta_0/partial x) minus the gradient in effective pressure (partial N_RC / partial x) consistent with equation 1b where theta_RC = rho_w*g theta_0 − N_RC. This is consistent with equation 3 from Kinglake and Ng (2013).*

In summary, I am really pleased that the authors have produced such an interesting and well-presented paper. I expect it will be well-read and useful and as I mentioned above it is interesting to think about its immediate implications for, among other things, the predictability of ice sheets.

*We could not have done it without you.*

---

## Editor Decision (ED1)

[revised manuscript text omitted]
 | $\text{M}^{0.47}\text{L}^{-0.47}\text{T}^{-1.94}$ | $3 \cdot 10^{-5}\ \text{Pa}^{b-a}\text{s}^{-1}$ |
| $A_L$ | Lake area | $\text{L}^2$ | Measured |
| $a$ | Constant of sediment deformation | - | 1.33 (Walder and Fowler, 1994) |
| $b$ | Constant of sediment deformation | - | 1.8 (Walder and Fowler, 1994) |
| $C_{VRC}$ | Rate of R-channel deformational closure | $\text{L}^2\text{T}^{-1}$ | Output by model |
| $C_{VCC}$ | Rate of viscous sediment closure | $\text{L}^2\text{T}^{-1}$ | Output by model |
| $\bar{c}$ | Sediment concentration in water column | - | Output by model |
| $c_S$ | Constant for roughness | $\text{M}^{-3}\text{L}^4\text{T}^5$ | $2 \cdot 10^{-20}\text{m s}^{-1}\ \text{Pa}^{-3}$ |
| $D_{CC}$ | Deposition rate | $\text{LT}^{-1}$ | Output by model |
| $d_{15}$ | Characteristic grain size | L | Input (see Table 2) |
| $d_{50}$ | Median grain size | L | Calculated from input |
| $E_{CC}$ | Erosion rate | $\text{LT}^{-1}$ | Output by model |
| $f_r$ | Hydraulic roughness | $\text{L}^{-2/3}\text{T}^2$ | $0.07\ \text{m}^{-2/3}\text{s}^2$ |
| $f_{CC}$ | Canal roughness parameter | $\text{L}^{-2/3}\text{T}^2$ | $0.07\ \text{m}^{-2/3}\text{s}^2$ |
| $F_L$ | Coefficient of lake flexure | - | Input (between 1 and 2 based on observations reported in Fricker and Scambos, 2009) |
| $g$ | Gravitational acceleration | $\text{LT}^{-2}$ | $9.81\ \text{ms}^{-2}$ |
| $h_S$ | Water layer thickness for sheet flow system | L | Measured |
| $h_i$ | Ice thickness | L | Measured |
| $K_{RC}$ | Glen's flow law parameter for ice | $\text{L}^3\text{T}^5\text{M}^{-3}$ | $10^{-24}\text{Pa}^{-3}\ \text{s}^{-1}$ (Kingslake and Ng, 2013) |
| $K_{S1}$ | Constant | - | 1.1 |
| $K_{T1}$ | Erosional constant | - | 0.1 |
| $K_{T2}$ | Constant of deposition | - | 6 |
| $k$ | Constant of sheet-conduit transfer | $\text{M}^{-1}\text{L}^3\text{T}$ | $10^{-9}\ \text{m}^2\text{s}^{-1}\text{Pa}^{-1}$ (Kingslake and Ng, 2013) |
| $k_h$ | Coefficient for partitioning of turbulent energy between heating water and melting surrounding ice | - | 0.309 (Hooke, 2005) |
| $L_h$ | Latent heat of fusion | $\text{L}^2\text{T}^{-2}$ | $333{,}500\ \text{J kg}^{-1}$ |

[revised manuscript text omitted]

---

## Author Response (AR2)

Dear Dr. Bingham and the "The Cryosphere" editorial staff,

Thank you all for the time consideration and energy devoted to our manuscript.

An itemized list of the changes made appears below:

Page 1, Lines 2, 4, 9 and 19:

"Ice  sheet" has thrice been changed to "ice-sheet".    "Ice stream" has been changed to "ice-stream."

Page 1 line 24:

"Data and" has been changed to "data, and";  "to precisely observe the ice surface" has been changed to "to observe the ice surface precisely"

Page 2 line 1:

"Radio echo sounding" deleted

Page 2, line 8:  "fast flowing" changed to "fast-flowing"

Page 2, Line 21:  "of longer" changed to "of the longer"

Page 2, Lines 24, 25, and 34:

"Ice  sheet" has been changed to "ice-sheet".   "Lake drainage" changed to "lake-drainage."  "Ice stream" has been changed to "ice-stream."

Page 2 Line 29:  "Complimentary" changed to "Complementary"

Page 3, Lines 2, 4, 5 and 19:

"Ice  sheet" has been changed to "ice-sheet".   "Lake drainage" changed to "lake-drainage."  "Ice stream" has been changed to "ice-stream."

Page 3 line 24:

"Antarctic ice sheet" changed to "Antarctic Ice Sheet"

Page 3 line 29:

"thermally eroded" changed to "eroded thermally"

Page 4 Line 10:

"glacial-dammed" changed to "ice-dammed"; comma added after citation

Page 4 Line 11:

"Timescale" changed to "timescales";  "days and" changed to "days, and"

Page 4 line 19:

"Simultaneously decreases" changed to "decreases simultaneously"

Page 4 Lines 25, 29, and 31

"Ice dammed" changed to "ice-dammed";  "pressure melting" changed to "pressure-melting"

Page 5 Line 2:  Comma inserted

Page 5 line 6:  Apostrophe removed

Page 5 Line 8:  acronym removed

Page 5 line 16:  "Best" changed to "better"

Page 6 Line 3:  "comes" changed to "come"

Page 6 Line 4:  "water budget" changed to "water-budget";  "lake inflow" changed to "lake-inflow";  Acronym now used

Page 6 Line 5:  second comparing deleted; "lake-volume" changed to "lake-volume" twice

Page 6 Line 16:  "Water system" changed to "water-system"

Page 6 Line 20:  "as" changed to "because the"

Page 7 line 7:  "Principle" changed to "principal"

Page 7 line 9:  hyphen removed from "well posed"; hyphen added in "subglacial-lake drainage"

Page 7 line 15:  "pressure melting point" changed to "pressure-melting point"

Page 7 line 20:  "viscous ice"  changed to "viscous-ice"

Page 9 line 2:  "to account" changed to "which accounts"

Page 10 Line 2:  hyphen added to "sediment-collapse"

Page 10 Line 4:  "Where" changed to "here" thus answering one of life's greatest philosophical questions.  Maybe next time it will be "somewhere else"

Page 11 Line 3:  "Flow path" is now "flow-path";  "width changed to "with"

Page 11 Line 4:  ":" changes to ","

Page 11 Line 7:  order of "to" and "either" reversed

Page 11 Line 15:  "Linked cavity" changed to "linked-cavity"

Page 11 Line 22:  "ice base"  changed to "ice-base"

Page 12 Line 4:  I got rid of "that"

Page 12 Line 21: "Ice  sheet" has been changed to "ice-sheet".

Page 12 Line 24:  "Implimentation" changed to "implementation"

Page 12 Line 27:  "Discharge level" changed to "discharge-level"

Page 13 Line 3:  "Water layer" changed to "water-layer"

Page 13 Line 10:  comma added

Page 13 Line 11:  "Water layer" changed to "water-layer"

Page 13 Line 13:  "Criteria" changed to "criterion"

Page 13 Line 16:  "newly formed"  changed to "a newly-formed"

Page 13 Line 23:  comma added

Page 14 Line 6:  "Sheet flow" changed to "sheet-flow"

Page 14 Line 7: "Courrant" changed to "Courant"

Page 14 Line 22:  "Lake drainage" changed to "lake-drainage."

Page 14 Line 23:  "Draining lakes" changed to "draining the lakes";

Page 14 Line 23:  now reads " . . . in lower Whillans and Mercer Ice Streams: see Figures 4b (for SLW), 4c (for SLC and SLM), and 4d (for SLE)."

Page 14 line 25:  "real domain" changed to "real-domain"

Page 14 Line 27:  "Lythe and Vaughan" changed to "Lythe et al."

Page 14 Line 28:  comma added.

Page 15 Line 10: "ice surface" is now "ice-surface";  "draw down" is now "drawdown"

Page 15 Line 14-16:  "Lake volume" is now "lake-volume"

Page 15 Line 20: "provided" changed to "proved"

Page 15 Line 20: "better quantifying"  is now "quantifying better"

Page 15 Line 20:  "flow path" is now "flow-path"

Page 15 Line 21:  "affected" is now "affect"

Page 15 Line 22:  "to both" is now "both to"

Page 15 Line 23:  "directly measure" is now "measure directly"

Page 15 Line 27:  'lake volume" changed to "lake-volume"

Page 15 Line 27:  "qualitative similar" is now "qualitatively similar"

Page 15 Line 28:  "Lake drainage" changed to "lake-drainage."

Page 15 Line 31: "models" changed to "model's"

Page 16 Line 2:  "Lake drainage" changed to "lake-drainage."

Page 16 Line 3:  "30 year" is now "30-year"

Page 16 Line 9:  "began" is now "begin"

Page 16 Line 11:  "an" added to read "show an ~ 4 year"

Page 16 Line 20:  "Lake drainage" changed to "lake-drainage."

Page 16 Line 24:  "Lake volume" changed to "lake-volume."

Page 16 Line 28:  Figure reference removed.

Page 16 Line 29:  "Decreased and" is now "decreased, and"

Page 16 Line 32:  comma added between "increased" and "however"

Page 17 Line 2: "then draw" changed to "then to draw"

Page 17 Line 3:  comma inserted between "level" and "however"

Page 17 Line 10:  "lake volume" changed to "lake-volume"

Page 17 Line 11:  "Using a constant" changed to "using constant"

Page 17 Line 20:  "Flow path" is now "flow-path"

Page 17 Line 25:  "distributed system" changed to "distributed-system"

Page 17 Line 28:  "of the" added between "portion" and "total"

Page 17 Line 29:  "to" changed to "and"

Page 17 Line 32:  "lake volume" changed to "lake-volume"

Page 18 Line 3:  "lake volume" changed to "lake-volume"

Page 18 Line 3:  "rise in falls" changed to "rises and falls"

Page 18 Line 4:  Hyphen removed

Page 18 Line 8:  "higher volume" is now "higher-volume"

Page 18 Line 17:  "Water was leaving" changed to "water leaving"

Page 19 Line 2:  "range as such" changed to "range, because such"

Page 19 Line 5:  Comma inserted between "conversely" and "parameter"

Page 19 Line 12:  "the" removed

Page 19 Line 17:  "a" changed to "an";  comma added between "environment" and "supporting"

Page 19 Line 18:  "no" changed to "not"

Page 19 Line 21:  "radar" changed to "RES"

Page 19 Line 24:  "Ice  sheet" has been changed to "ice-sheet"

Page 19 line 26:  "Siple Coast style lake drainage" changed to "Siple-Coast-style lake-drainage"

Page 20 line 2:  comma inserted between "deformation" and "should"

Page 20 Line 8: "observations" deleted

Page 20 Line 9: "radar" changed to "RES"

Page 20 Line 11: "to effectively constrain our model" is now "to constrain our model effectively"

Page 20 Line 18: "system to eventually come" changed to "system eventually to come"

Page 20 Line 19: "increased" changed to "increases"

Page 20 Line 20: "related" changed to "relates"

Page 20 Line 23: "the a " changed to "a"

Page 20 Line 29: "work however" changed to " work, however"

Page 21 Line 2: "is a sufficient" changed to "acts as a"

Page 21 Line 7: "ice marginal" changed to "ice-marginal"

Page 21 Line 13: "radar" changed to "RES"

Page 21 Line 13: "More traditional radar sounding technology" changed to "RES"

Page 21 Line 16: "sediment" changed to "sediments"

Page 21 Line 18: "radar sounding" changed to "RES"

Page 21 Line 21: "of" changed to "between"

Page 21 Line 28: "Lake drainage" changed to "lake-drainage."

Page 21 Line 28: "to fully understand" changed to "to understand fully"

Page 21 Line 30:  "Ice  sheet" has been changed to "ice-sheet"

Page 22 Line 1:  "ice bed" changed to "ice – bed"

Page 22 Line 3:  "from" changed to "of"

Page 22 Line 7:  "velocity" changed to "speed"

Page 22 Line 16 and 18:  "Ice  sheet" has been changed to "ice-sheet"

Page 22 Line 23:  "Ice stream" has been changed to "ice-stream"

Page 22 Line 24:  "effective pressure" changed to "effective-pressure"

Page 22 Line 25:  "correlated" changed to "correlates"

Page 22 Line 31:  "that" changed to "the"

Page 22 Line 32:  comma added between "intriguingly" and "recent"; "WIS" expanded into "Whillans Ice Stream"

Page 23 Line 1:  "to" added between "necessary" and "create"

Page 23 Line 2:  Hodson added to reference list.

Page 23 Line 3 and 5:  "radar sounding" is now "radio-echo sounding"

Page 29:  "Satellite detected" is now "satellite-detected"; "radar sounding" is now "radio-echo sounding"; and "Ice Streams" is now capitalized.

Page 31:  spelling of "floatation" changed to "flotation"

Page 32: ice base changed to "ice base"; "figures" changed to "figure"

Page 33:  "gray" changed to "grey"

Page 34:  from "(Carter et al., 2013)" changed to from "Carter et al., (2013)"